# A Bi-Level Framework for Learning to Solve Combinatorial Optimization on Graphs

**Runzhong Wang**[1][*]   **Zhigang Hua**[2]   **Gan Liu**[2]   **Jiayi Zhang**[1]   **Junchi Yan**[1]([✉])[†]
**Feng Qi**[2]   **Shuang Yang**[2]   **Jun Zhou**[2]   **Xiaokang Yang**[1]
[1] Department of CSE and MoE Key Lab of AI, Shanghai Jiao Tong University   [2] Ant Group
{runzhong.wang,zhangjiayirr,yanjunchi,xkyang}@sjtu.edu.cn
{z.hua,liugan.lg,feng.qi,shuang.yang,jun.zhoujun}@antgroup.com

## Abstract

Combinatorial Optimization (CO) has been a long-standing challenging research topic featured by its NP-hard nature. Traditionally such problems are approximately solved with heuristic algorithms which are usually fast but may sacrifice the solution quality. Currently, machine learning for combinatorial optimization (MLCO) has become a trending research topic, but most existing MLCO methods treat CO as a single-level optimization by directly learning the end-to-end solutions, which are hard to scale up and mostly limited by the capacity of ML models given the high complexity of CO. In this paper, we propose a hybrid approach to combine the best of the two worlds, in which a bi-level framework is developed with an upper-level learning method to optimize the graph (e.g. add, delete or modify edges in a graph), fused with a lower-level heuristic algorithm solving on the optimized graph. Such a bi-level approach simplifies the learning on the original hard CO and can effectively mitigate the demand for model capacity. The experiments and results on several popular CO problems like Directed Acyclic Graph scheduling, Graph Edit Distance and Hamiltonian Cycle Problem show its effectiveness over manually designed heuristics and single-level learning methods. Code available at https://github.com/Thinklab-SJTU/PPO-BiHyb.

## 1 Introduction

Combinatorial Optimization (CO) is a family of long-standing optimization problems. A large portion of CO problems is NP-hard due to the combinatorial nature, raising challenges for traditional (exact) solvers on even medium-sized problems. Heuristic algorithms are often adopted to approximately solve CO problems within an acceptable time, and there is a growing trend adopting modern data-driven approaches to solve CO problems that achieve better and faster results [30].

The major line of works solving CO with machine learning (ML) is single-level [8, 25, 29, 30, 39, 41, 57, 59, 64], where the prediction of ML module lies in the solution space, assuming the model has enough capacity learning the input-output mapping of the CO problem. However, achieving such an assumption is non-trivial, leading to the following two aspects of challenges. On the one hand, it is challenging to design a model with enough capacity with limited computational resources, and existing models are usually tailored for specific problems which require heavy trail-and-error [25, 57, 59]. On the other hand, training such a heavy model requires either supervision from high-quality labels [31, 57, 60] which are infeasible to obtain for large-sized problems due to the NP-hard nature, or reinforcement learning (RL) [8, 30, 38, 39] which might be unstable due to the challenges of large action space and sparse reward especially for large-sized problems [52].

---

[*]Part of the work was done while the first author was working as an intern at Ant Group.
[†]Junchi Yan is the corresponding author.

35th Conference on Neural Information Processing Systems (NeurIPS 2021), virtual.

An alternative approach resorts to a hybrid machine learning and traditional optimization pipeline [11, 18, 26, 31, 51, 60, 61] hoping to utilize the power of traditional optimization methods. However, designing a general hybrid MLCO approach is still non-trivial, as existing methods [11, 60] usually require domain-specific knowledge for the model design. It is again challenging to obtain high-quality supervision labels, and existing methods are based on either problem-specific surrogate labels [18, 31, 60], or learned with RL while the challenges of RL still exist [11, 51].

In this paper, we propose a general hybrid MLCO approach over graphs. We first reduce the complexity of deep learning model by reformulating the original CO into a bi-level optimization, whose objective is to minimize the long-term upper-level objective by optimizing the graph structure, and the lower-level problem is handled by an existing heuristic. We resort to RL and the traditional heuristic can be absorbed as part of the environment, and it is shown that the sparse reward issue is mitigated for the resulting RL problem compared to previous RL-based methods [8, 30, 38, 39]. Specifically, our model is built with standard building blocks: the input graph is encoded by Graph Convolutional Network (GCN) [32], and the actor and critic modules are based on ResNet blocks [22] and attention models [55]. All modules are learned with the Proximal Policy Optimization (PPO) algorithm [48]. **The contributions of this paper include:**

• To combine the best of the two worlds, we propose a general hybrid approach that integrates traditional heuristic solvers with machine learning algorithm.

• We propose a bi-level optimization formulation for learning to solve CO on graphs. The upper-level optimization adopts a reinforcement learning agent to adaptively modify the graphs, while the lower-level optimization involves traditional learning-free heuristics to solve combinatorial optimization tasks on the modified graphs. Our approach does not require ground truth labels.

• The experiments for graphs up to thousands of nodes on several popular tasks such as Directed Acyclic Graph scheduling (DAG scheduling), Graph Edit Distance (GED), and Hamiltonian Cycle Problem (HCP) show that our method notably surpasses both traditional learning-free heuristics and single-level learning method. Our method generalizes well to graphs of different sizes while having comparable overhead to the single-level learning methods.

## 2   Related Work

Here we discuss related works listed in recent surveys [4, 62, 63]. We present their methodologies and compare with our method.

**Learning the end-to-end solution as a sequence.** The major line of existing MLCO works mainly focus on tackling the problem end-to-end by predicting a sequence of solutions [25, 29, 30, 35, 39, 41, 57, 59, 64]. The pioneering work [57] is designated for traveling salesman problem (TSP), whereby the sequence-to-sequence Pointer Network (PtrNet) model is learned with supervised learning. In [30], a graph-to-sequence framework is proposed with graph embedding [10] and Q-learning [40] for general CO over graphs, and this general framework inspires the major line of MLCO works with applications to DAG scheduling [39], graph matching [36], and job shop scheduling [64]. [35] extends [30] by problem reduction and supervised learning-based tree search. The framework in [30] is treated as the RL baseline in this paper. Our approach differs from these single-level end-to-end RL methods [30, 36, 38, 39], which lack the flexibility to borrow the power of traditional methods and often suffer from the fundamental issues in RL: sparse reward and large action space.

**Learning to rewrite end-to-end solutions.** Another line of end-to-end learning works predicts rewriting strategies of an existing solution [8, 38], where the model predictions also lie in the solution space. The agents learn to improve an existing solution in a decision sequence, and problems like job scheduling, expression simplification and routing problems have been studied. These learning-based local search heuristics also fall within the single-level paradigm, while our method incorporates another optimization over the graph structure. The sparse reward issue also exists for [8, 38], as a long episode is usually required for searching for a satisfying result.

**One-shot unsupervised end-to-end learning.** There is also a growing trend applying one-shot unsupervised methods for the input-output mapping of CO, with maximum clique and graph cut solved by [29], and quadratic assignment problem solved by [59]. However, the generalization ability of these methods is still an open question, and complicated constraints are often non-trivial to be

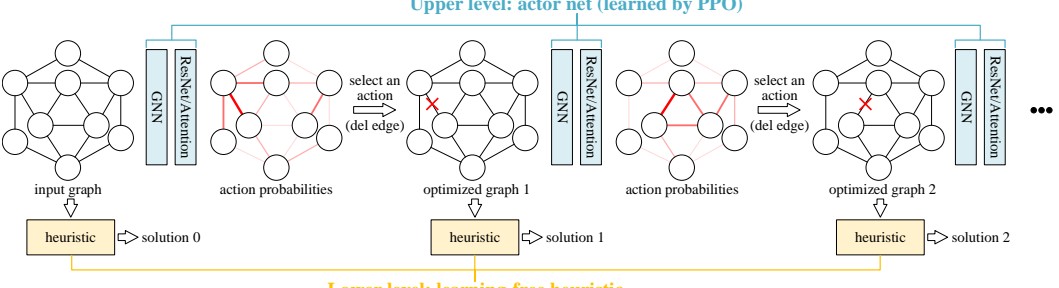

Figure 1: An overview of our bi-level hybrid MLCO solver. The graph structure is optimized at the upper level by an RL agent, and the optimized graphs are solved by heuristics at the lower level. The actions can be any modifications of the edges (i.e. adding, deleting edges and modifying edge attributes), and edge deletion is presented in this example.

encoded in the network's prediction. Moreover, such a one-shot end-to-end network often calls for much higher model capacity compared with our multi-round alternating optimization paradigm.

**Hybrid of machine learning and traditional solvers.** Different from learning the solution end-to-end, researchers also propose hybrid machine learning and traditional solver approaches. ML modules are studied as sub-routines for traditional solvers, especially predicting branching strategies for branch-and-bound with either supervised learning [18, 31] or reinforcement learning [51]. In [60], the heuristic routine in A* algorithm is replaced by graph neural network to solve graph edit distance. However, these methods are tailored for special problems, and our aim is to develop a more general framework, where the learning part and heuristic module are two peers alternatively performed.

**Bi-level optimization.** Our method is based on bi-level optimization, which is a family of optimization problems where a lower-level optimization is nested inside an upper-level optimization. Bi-level optimization is in general NP-hard [27, 56], and the applications of bi-level optimization can be found ranging from multi-player games [27] to vision tasks [37], and there is a loosely relevant attempt [2] adopting supervised learning to solve a bi-level optimization for transportation.

## 3   Our Approach

In this paper, we propose a **Bi**-level **Hyb**rid (BiHyb) machine learning and traditional heuristic approach. Sec. 3.1 shows both single-level and bi-level formulations of CO, and Sec. 3.2 shows the RL approach to the bi-level CO.

### 3.1   Bi-level Reformulation of Combinatorial Optimization

Without loss of generality, we consider the classic single-level CO with a single graph $\mathcal{G}$ as:

$$\min_{\mathbf{x}} f(\mathbf{x}|\mathcal{G}) \qquad s.t. \quad h_i(\mathbf{x}, \mathcal{G}) \leq 0, \text{for } i = 1...I \qquad (1)$$

where $\mathbf{x}$ denotes the decision variable (i.e. solution), $f(\mathbf{x}|\mathcal{G})$ denotes the objective function given input graph $\mathcal{G}$ and $h_i(\mathbf{x}, \mathcal{G}) \leq 0$ represents the set of constraints. For example, in DAG scheduling, the constraints enforce that the solution $\mathbf{x}$, i.e. the execution order of the DAG job nodes, lies in the feasible space and does not conflict the topological dependency structure of $\mathcal{G}$. The popular framework of existing MLCO methods regards Eq. 1 as a straight-forward end-to-end learning task, and various training methods have been developed, including: 1) supervised learning [41, 57] obtaining training labels by solving small-scaled Eq. 1 with traditional solvers, however it is nearly infeasible to solve larger NP-hard problems; 2) unsupervised learning [29, 59] adopting the continuous relaxation of Eq. 1 as the learning objective, but existing methods face challenges when dealing with complicated constraints; and 3) reinforcement learning [8, 30] by predicting $\mathbf{x}$ sequentially, but the reward signal is unavailable until $\mathbf{x}$ reaches a complete solution, leading to the sparse reward issue.

To ease the challenges introduced by the single-level formulation, we resort to the classic idea of modifying the original problem to aid problem solving, e.g. adding cutting planes for integer programming [19, 54]. Our proposed framework is capable of handling CO on graphs if all constraints can be encoded by the graph structure, and our motivation is described by the following hypothesis:

**Algorithm 1: Policy Roll-out for Bi-level Learning of Hybrid MLCO Solver (BiHyb)**

**Input:** Original graph $\mathcal{G}$; Max number of actions $K$.

1   $\mathcal{G}^0 \leftarrow \mathcal{G}$;
2   **for** $k \leftarrow 0 \dots (K-1)$ **do**
3     Predict $P(\mathbf{a}_1), P(\mathbf{a}_2|\mathbf{a}_1)$ on $\mathcal{G}^k$ and sample $\mathbf{a}_1, \mathbf{a}_2$; # upper-level optimization for learning
4     $\mathcal{G}^{k+1} \leftarrow$ add, delete or modify the edge $(\mathbf{a}_1, \mathbf{a}_2)$ in $\mathcal{G}^k$; # state-transition
5     $\mathbf{x}^{k+1} \leftarrow$ solve $\arg\min_{\mathbf{x}^{k+1}} f(\mathbf{x}^{k+1}|\mathcal{G}^{k+1})$ by heuristic algorithm; # lower-level optimization
6     $r_k \leftarrow f(\mathbf{x}^k|\mathcal{G}) - f(\mathbf{x}^{k+1}|\mathcal{G})$; # reward

**Output:** A list of rewards $\{r_0 \dots r_{K-1}\}$.

The optimal solution $\mathbf{x}^*$ to $\mathcal{G}$ can be acquired by modifying $\mathcal{G}$. And we show the feasibility of this hypothesis for a family of problems by introducing the following proposition:

**Proposition.** *We define $\mathbb{G}$ as the set of all graphs that can be modified from $\mathcal{G}$, and $\mathbb{X}$ as the set of all feasible solutions of $\mathcal{G}$. If the heuristic algorithm is a surjection from $\mathbb{G}$ to $\mathbb{X}$, for $\mathcal{G}$ and its optimal solution $\mathbf{x}^*$, there must exist $\mathcal{G}^* \in \mathbb{G}$, such that $\mathbf{x}^*$ is the output of the heuristic by solving $\mathcal{G}^*$.*

*Proof.* By the definition of surjection, since $\mathbf{x}^* \in \mathbb{X}$, there must exist at least one graph $\mathcal{G}^* \in \mathbb{G}$ such that $\mathbf{x}^*$ is the output of the heuristic algorithm by solving $\mathcal{G}^*$. $\qquad\square$

We take *DAG scheduling* as an example to clarify this Proposition. Without loss of generality, we define processing the nodes 1 to $n$ sequentially as a feasible solution. Then, we can modify the graph as follows: if the edge connecting $i$ to $i+1$ does not exist, it is added. After adding all edges from 1 to $n$, processing the nodes from 1 to $n$ sequentially is the only feasible solution, which is also the output of any heuristic algorithm. The above construction method applies for all solutions in $\mathbb{X}$.

**Some Remarks.** Our hypothesis and proposition provide the theoretical grounding of developing graph-modification methods to tackle combinatorial problems. It is worth noting that $\mathcal{G}^*$ only suggests that graph modification is a promising direction and finding $\mathcal{G}^*$ given $\mathcal{G}$ is usually NP-hard in practice. In this paper, we propose to improve the solving quality for heuristic algorithms by finding optimized (not necessarily optimal) graphs by learning based on the bi-level reformulation of the original single-level problem. Due to practical reasons, we restrict the max number of modifications.

In the bi-level reformulation of the original single-level problem, an optimized graph $\mathcal{G}'$ is introduced:

$$\min_{\mathbf{x}', \mathcal{G}'} \ f(\mathbf{x}'|\mathcal{G}) \qquad s.t. \quad H_j(\mathcal{G}', \mathcal{G}) \leq 0, \text{for } j = 1 \dots J$$
$$\mathbf{x}' \in \arg\min_{\mathbf{x}'} \{ f(\mathbf{x}'|\mathcal{G}') : h_i(\mathbf{x}', \mathcal{G}') \leq 0, \text{for } i = 1 \dots I \} \tag{2}$$

where $f(\mathbf{x}'|\mathcal{G}), f(\mathbf{x}'|\mathcal{G}')$ are the objectives for upper- and lower-level problems, respectively. The lower-level problem is the CO given the optimized graph $\mathcal{G}'$, which is solved by a heuristic algorithm. The solved decision variable $\mathbf{x}'$ is further fed to the upper-level problem, whose objective $f(\mathbf{x}'|\mathcal{G})$ denotes the original CO objective computed by $\mathbf{x}'$. The upper-level constraints $H_j(\mathcal{G}', \mathcal{G}) \leq 0$ ensure that the feasible space of $\mathcal{G}'$ is a subset of $\mathcal{G}$, and $\mathcal{G}'$ has at most $K$ modification steps from $\mathcal{G}$. The upper-level problem optimizes $\mathcal{G}'$ by an RL agent, by regarding Eq. 2 as the environment.

### 3.2   Reinforcement Learning Algorithm

We resort to reinforcement learning to optimize $\mathcal{G}'$ in Eq. 2, which can be viewed as a data-driven embodiment of the classic bi-level optimization method by solving two levels of problems alternatively [53]. In Sec. 3.2.1 we present the Markov Decision Process (MDP) formulation of the bi-level optimization in Eq. 2, in Sec. 3.2.2 we describe the PPO learning algorithm in our approach.

#### 3.2.1   The MDP Formulation

Eq. 2 is treated as the learning objective and optimized by RL in a data-driven manner. In this section, we discuss the Markov Decision Process (MDP) formulation for adopting RL to this bi-level optimization problem. The policy roll-out steps are summarized in Alg. 1. In the following, $\mathcal{G}^0$ equals $\mathcal{G}$ meaning the original graph, and $\mathcal{G}^k (k \neq 0)$ equals $\mathcal{G}'$ meaning the modified graph after action $k$.

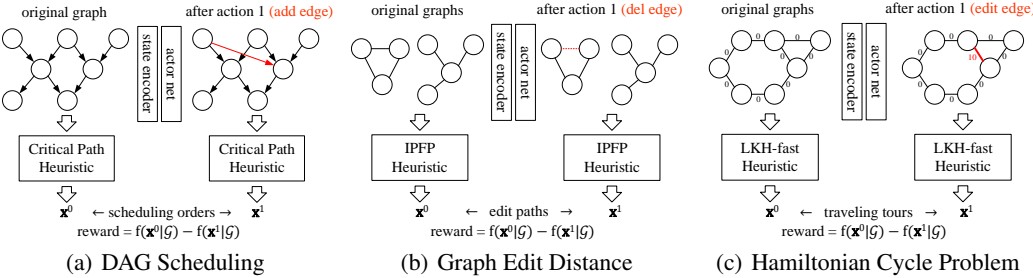

| (a) DAG Scheduling | (b) Graph Edit Distance | (c) Hamiltonian Cycle Problem |

Figure 2: Illustration of the state, action and reward for all problems discussed in this paper.

**State.** The current graph $\mathcal{G}^k$ is treated as the *state*, whose nodes and edges encode both the problem input and the current constraints. The starting *state* $\mathcal{G}^0$ represents the original CO problem.

**Action.** The *action* is defined as adding, removing or modifying an edge in $\mathcal{G}^k$. Since there are at most $m^2$ edges if $\mathcal{G}^k$ has $m$ nodes, we shrink the action space to $O(m)$ by decomposing the edge selection as two steps: firstly selecting the starting node, and then selecting the ending node.

**State transition.** After taking an *action*, $\mathcal{G}^k$ transforms to $\mathcal{G}^{k+1}$ with one edge modified. The new graph $\mathcal{G}^{k+1}$ is regarded as the new *state* and is adopted for *reward* computation. The episode ends when it reaches the max number of actions $K$. In our implementation, we empirically set $K \leq 20$ for graphs up to thousands of nodes, therefore the sparse reward issue is mitigated compared to single-level RL methods (20 actions v.s. 1000+ actions per episode).

**Reward.** The new graph $\mathcal{G}^{k+1}$ results in a modified lower-level optimization problem whose objective becomes $f(\mathbf{x}^{k+1}|\mathcal{G}^{k+1})$, and $\mathbf{x}^{k+1}$ is solved by an existing heuristic. The *reward* is computed as the decrease of upper-level objective function given $\mathbf{x}^{k+1}$: reward $= f(\mathbf{x}^k|\mathcal{G}) - f(\mathbf{x}^{k+1}|\mathcal{G})$.

### 3.2.2 Proximal Policy Optimization (PPO)

We resort to the popular Proximal Policy Optimization (PPO) [48] as the RL framework. PPO is the simplified version of Trust Region Policy Optimization (TRPO) [47] where the model update is restricted within a "trust region" to avoid model collapse. PPO is easier to implement than TRPO with comparative performance [48], maximizing the objective: $J(\theta) = \min(r_\theta \cdot A, \ clip(r_\theta, 1-\epsilon, 1+\epsilon) \cdot A)$ where $r_\theta$ is the importance sampling ratio parameterized by model parameter $\theta$, $A$ is the advantage value computed by the discounted accumulative reward minus the critic network prediction, and $\epsilon$ is a hyperparameter controlling the boundary of trust region. Some common policy-gradient training tricks are also adopted: we normalize the accumulated rewards during model update, and we add an entropy regularizer to encourage exploration beyond local optimums.

## 4 Experiments and Case Studies

We show the implementations and experiments on three challenging CO problems: DAG scheduling in Sec. 4.1, graph edit distance (GED) in Sec. 4.2, and Hamiltonian cycle problem (HCP) in Sec. 4.3. Our bi-level RL method PPO-BiHyb is compared with learning-free heuristics, and a single-level RL peer method namely PPO-Single following the general framework [30], which also covers the majority of RL-based methods [15, 25, 33, 36, 39, 64]. We also implement Random-BiHyb which performs random graph modification under our bi-level optimization framework. The model capacity and training/inference time of PPO-Single are kept in line with PPO-BiHyb for fair comparison.

### 4.1 Case 1: DAG Scheduling

The Directed Acyclic Graph (DAG) is the natural representation of real-world jobs with dependency, and the DAG scheduling problem is the abstraction of parallel job scheduling in computer clusters: each node represents a computation job with a running time and a resource requirement, and the node may have several parents and children representing data dependency. The cluster has limited total resources, and jobs can be executed in parallel if there are enough resources and the concurrent jobs do not have data dependency. Such an optimization problem is in general NP-hard [14], and the objective is to minimize the makespan of all jobs, i.e. finish all jobs as soon as possible.

#### 4.1.1 Implementation Components

**MDP Formulation.** As shown in Fig. 2(a), *state* is the current DAG $\mathcal{G}^k$, *action* is defined as adding an edge to $\mathcal{G}^k$, resulting in a new DAG $\mathcal{G}^{k+1}$. The added edge enforces additional constraints such that the decision space is cut down to elevate the heuristic's performance. Here $\mathbf{x}$ represents the scheduled execution order, and the Critical Path heuristic is adopted on $\mathcal{G}^{k+1}$ to compute $\mathbf{x}^{k+1}$. The *reward* is computed by subtracting the previous makespan by the new makespan: $f(\mathbf{x}^k|\mathcal{G}) - f(\mathbf{x}^{k+1}|\mathcal{G})$.

**State Encoder.** We adopt GCN [32] to encode the state represented by DAG. Considering the structure of DAG, we design two GCNs: the first GCN processes the original DAG, and the second GCN processes the DAG with all edges reversed. The node embeddings from two GCN modules are concatenated, and an attention pooling layer is adopted to extract a graph-level embedding.

$$\mathbf{n} = \left[ \text{GCN}_1(\mathcal{G}^k) \,||\, \text{GCN}_2(\text{reverse}(\mathcal{G}^k)) \right], \ \mathbf{g} = \text{Att}(\mathbf{n}) \tag{3}$$

where $\mathbf{n}$ (# of nodes $\times$ embedding dim) is the node embedding, $[\cdot \,||\, \cdot]$ means concatenation.

**Actor Net.** To reduce action space, the edge selection is decomposed into two steps: first selecting the starting node and then the ending node. The action probabilities of selecting the starting and ending nodes are predicted by two independent 3-layer ResNets blocks [22], respectively, and the input of the second ResNet contains additionally the feature vector of the selected starting node.

$$P(\mathbf{a}_1) = \text{softmax}(\text{ResNet}_1([\mathbf{n} \,||\, \mathbf{g}])), \ P(\mathbf{a}_2|\mathbf{a}_1) = \text{softmax}(\text{ResNet}_2([\mathbf{n} \,||\, \mathbf{n}[\mathbf{a}_1] \,||\, \mathbf{g}])) \tag{4}$$

the subscript of $\mathbf{n}[\mathbf{a}_1]$ denotes the embedding for a node $\mathbf{a}_1$. For training, we sample $\mathbf{a}_1, \mathbf{a}_2$ according to their probabilities $P(\mathbf{a}_1), P(\mathbf{a}_2|\mathbf{a}_1)$ respectively. For testing, beam search is performed with a width of 3: actions with top-3 highest probabilities are searched, and all searched actions are evaluated by the improvement of makespan, and only the top-3 actions are maintained for the next search step.

**Critic Net.** It is built by max pooling over all node features, and the pooled feature is concatenated with the graph feature from state encoder, and finally processed by another ResNet for value prediction.

$$\widetilde{V}(\mathcal{G}^k) = \text{ResNet}_3\left([\text{maxpool}(\mathbf{n}) \,||\, \mathbf{g}]\right) \tag{5}$$

**Heuristic Methods.** Solving DAG scheduling with hundreds of nodes is nearly infeasible for existing commercial solvers, and real-world scheduling problems are usually tackled by fast heuristics e.g. *Shortest Job First* which schedules the shortest job greedily and *Critical Path* algorithm which prioritizes jobs on the critical path. We also consider the novel *Tetris* scheduling [20] where jobs are arranged as a Tetris game on the two-dimension space of makespan and resource. Since *Critical Path* is empirically effective, we use it as the lower level optimization algorithm in our PPO-BiHyb learning method.

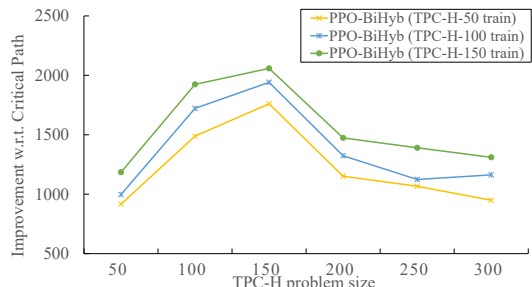

Figure 3: Generalization result on TPC-H dataset.

**Learning Methods.** There are some efforts to train a scheduler in data centers with RL [39] and learning a job shop scheduler with RL [64]. They can be viewed as embodiments with tailored techniques for specific problems based on the end-to-end single-level pipeline [30]. Since the problem settings in [39, 64] are different with ours, we re-implement the single-level RL baseline PPO-Single following [30], where the model details are in line with our PPO-BiHyb. To validate the effectiveness of PPO, we also compare Random-BiHyb which performs random search instead of PPO when modifying the graph, and its search steps are equal to PPO-BiHyb.

#### 4.1.2 Experimental Results

Results are reported for scheduling jobs from TPC-H dataset[3], which is composed of business-oriented queries and concurrent data modifications, represented by Directed Acyclic Graphs (DAGs). Each DAG represents a unique computation job, and each node in DAG has two properties: execution time and resource requirement. The DAG in TPC-H dataset has 9.18 nodes in average, and the smallest job has 2 nodes and the largest one has 18 nodes. The average resource requirement is

---

[3]http://tpc.org/tpch/default5.asp

Table 1: Experimental results on DAG scheduling problems from TPC-H dataset with the average number of nodes reported in brackets. The dataset name TPC-H-X means X jobs are jointly scheduled. The upper half is learning-free heuristics, and the lower half is RL-based methods where PPO-Single can be viewed as our implementation of the peer single-level RL methods [39, 64]. Note "objective" means the objective score i.e. the total makespan time (in seconds, the smaller the better) to finish all jobs, and "relative" is computed by the solved makespan time w.r.t. Critical Path heuristic.

| dataset / method | TPC-H-50 (#nodes=467.2) objective ↓ | relative ↓ | TPC-H-100 (#nodes=929.8) objective ↓ | relative ↓ | TPC-H-150 (#nodes=1384.5) objective ↓ | relative ↓ |
|---|---|---|---|---|---|---|
| Shortest Job First | 12818±2214 | 30.5% | 19503±3260 | 15.3% | 27409±2748 | 12.2% |
| Tetris [20] | 12113±1398 | 23.3% | 18291±2223 | 8.1% | 25325±2842 | 3.7% |
| **Critical Path** | 9821±1176 | 0.0% | 16914±2499 | 0.0% | 24429±2484 | 0.0% |
| PPO-Single [30, 39] | 10578±2092 | 7.7% | 17282±3821 | 2.2% | 24822±2707 | 1.6% |
| Random-BiHyb | 9270±1143 | -5.6% | 15580±2409 | -7.9% | 22930±2408 | -6.1% |
| **PPO-BiHyb (ours)** | **8906±922** | **-9.3%** | **15193±2275** | **-10.2%** | **22371±2538** | **-8.4%** |

125.8, and the minimum is 1 and the maximum is 593. The average task duration is 1127.2 sec, and the minimum is 16.3 sec and the maximum is 4964.5 sec. We jointly schedule multiple DAGs and assume there are 6000 total resources (i.e. the heaviest job node consumes around 10% of total resources), to reflect real-world business scenarios. We build 50 training and 10 testing samples for all experiments, where each sample is composed of a number of DAGs (e.g. 50, 100, 150) which are randomly sampled from the TPC-H dataset with fixed random seed.

Table 1 reports the result on scheduling jobs from TPC-H dataset, where our PPO-BiHyb outperforms learning-free heuristics and the single level learning method. We control the beam search width of PPO-Single as 10 allowing its inference time to be $10\times$ longer than ours, but its performance is still inferior to the heuristic Critical Path method. Random-BiHyb can improve the lower-level heuristic algorithm but its performance is still inferior to PPO-BiHyb, suggesting the effectiveness of both our bi-level optimization framework and PPO training. We also study the generalization of our learning approach, which is critically important for real-world applications. As shown in Fig. 3, our models are learned with TPC-H-50/100/150 datasets, and we report their test results on the unseen datasets ranging from TPC-H-50 to 300 at 50 interval. The plotted results are the improvement of makespan w.r.t. Critical Path. All our learning models generalize well with unseen data, even on larger problems with up to 300 DAGs (~2800 nodes). The generalization results of PPO-Single are omitted because they are consistently inferior to the Critical Path heuristic. The reason that our method generalizes soundly, in our analysis, is because learning the graph-optimization strategy is easier compared to directly learning the solution, and the strength of heuristics is exploited with our approach.

## 4.2 Case 2: Graph Edit Distance

The graph edit distance (GED) problem is NP-hard [1] and can be readily used in modeling a family of pattern recognition tasks requiring to measure the similarity between graphs, with applications to drug discovery [43], malware detection [34] and scene graph edition [7]. Given two graphs, the objective is finding the cheapest edit path from one graph to the other, and the costs of edit operations are defined by problem-specific distance metrics. For example, for molecule graphs, we may define an edit cost between different atom types (i.e. different node types), and also an edit cost for creating/removing a chemical bound (i.e. adding/removing an edge).

### 4.2.1 Implementation Components

**Bi-level Optimization Formulation.** Since GED involves two graphs, the formulation of the bi-level optimization of GED is a generalization from Eq. 2:

$$\min_{\mathbf{x}', \mathcal{G}_1'} f(\mathbf{x}'|\mathcal{G}_1, \mathcal{G}_2) \qquad s.t. \quad H_j(\mathcal{G}_1', \mathcal{G}_1) \leq 0, \text{for } j = 1...J$$
$$\mathbf{x}' \in \arg\min_{\mathbf{x}'} \{f(\mathbf{x}'|\mathcal{G}_1', \mathcal{G}_2) : h_i(\mathbf{x}', \mathcal{G}_1', \mathcal{G}_2) \leq 0, \text{for } i = 1...I\} \qquad (6)$$

where $f(\mathbf{x}'|\mathcal{G}_1, \mathcal{G}_2)$ is the upper-level objective: the graph edit cost given $\mathcal{G}_1, \mathcal{G}_2$ and $\mathbf{x}'$. The decision variable $\mathbf{x}'$ encodes the node editions from $\mathcal{G}_1$ to $\mathcal{G}_2$, based on which the edge editions can be induced. The upper-level constraints $H_j(\mathcal{G}_1', \mathcal{G}_1) \leq 0$ ensure that $\mathcal{G}_1'$ has at most $K$ modification steps from $\mathcal{G}_1$. The lower-level GED problem is defined between $\mathcal{G}_2$ and the modified $\mathcal{G}_1'$, where $f(\mathbf{x}'|\mathcal{G}_1', \mathcal{G}_2)$ is the lower-level objective, and $h_i(\mathbf{x}', \mathcal{G}_1', \mathcal{G}_2) \leq 0$ are the constraints of the lower-level GED.

**MDP Formulation.** As shown in Fig. 2(b), since there are two graphs, we discriminate the graphs by subscripts $\mathcal{G}_1, \mathcal{G}_2$, and the MDP formulation is a generalized version of Sec. 3.2.1. More specifically, the RL agent is restricted to modifying $\mathcal{G}_1$ and keeping $\mathcal{G}_2$ fixed. The *state* is defined as the current version of first graph $\mathcal{G}_1^k$, and *action* is defined as adding or removing an edge from $\mathcal{G}_1^k$. The new graph $\mathcal{G}_1^{k+1}$ is designed to better align with $\mathcal{G}_2$, where the node-to-node alignment is encoded by the decision variable $\mathbf{x}^k$ which is obtained by solving $f(\mathbf{x}^k | \mathcal{G}_1^k, \mathcal{G}_2)$ with IPFP heuristic. The upper-level objective $f(\mathbf{x}^k | \mathcal{G}_1, \mathcal{G}_2)$ is computed as the edit distance between the original graphs $\mathcal{G}_1, \mathcal{G}_2$ using $\mathbf{x}^k$.

**State Encoder.** The state encoder is built with GCN [32] to aggregate graph features and Sinkhorn-Knopp (SK) network [49] for propagation across two graphs. Following [58], the SK module accepts a similarity matrix (by inner-product of $\mathbf{n}_1$ and $\mathbf{n}_2$), and outputs a doubly-stochastic matrix which can be utilized as the cross-graph propagation weights. As the action is taken on the first graph, we compute the difference of the node features of graph 1 and the propagated features from graph 2 through SK net. Similar to the DAG model, graph-level features are obtained via attention pooling.

$$\mathbf{n}_1 = \text{GCN}(\mathcal{G}_1^k), \ \mathbf{n}_2 = \text{GCN}(\mathcal{G}_2), \ \mathbf{n} = \mathbf{n}_1 - \text{SK}(\mathbf{n}_1 \mathbf{n}_2^\top) \cdot \mathbf{n}_2; \ \mathbf{g}_1 = \text{Att}(\mathbf{n}_1), \ \mathbf{g}_2 = \text{Att}(\mathbf{n}_2) \quad (7)$$

**Actor Net.** The action of selecting an edge to add or delete is also decomposed by two node-selection steps. The starting node selection is predicted by 3-layer ResNet module, and the ending node selection is predicted by an attention query. The edge is deleted if it already exists, or added otherwise. The beam width is set as 3 during evaluation.

$$P(\mathbf{a}_1) = \text{softmax}\left(\text{ResNet}(\mathbf{n})\right), \ P(\mathbf{a}_2 | \mathbf{a}_1) = \text{softmax}(\mathbf{n} \cdot \tanh(\text{Linear}\left(\mathbf{n}[\mathbf{a}_1]\right))^\top) \quad (8)$$

**Critic Net.** We follow the graph-wise similarity learning method [3] to implement the critic net, where the graph-level features are processed by a neural tensor network (NTN) [50] followed by 2 fully-connected regression layers whose output is one-dimensional:

$$\widetilde{V}(\mathcal{G}_1^k, \mathcal{G}_2) = \text{fc}\left(\text{NTN}\left(\mathbf{g}_1, \mathbf{g}_2\right)\right) \quad (9)$$

**Heuristic Methods.** Based on the comprehensive evaluation on different GED heuristics by [5], we select 4 best-performing heuristics: *Hungarian* [44] which simplifies the original problem to a bipartite matching problem, *RRWM* [9] which solves GED via relaxed quadratic programming, *Hungarian-Search* [45] which is a search algorithm guided by Hungarian heuristic, and *IPFP* [6] which combines searching and quadratic programming. We empirically find *IPFP* best performs among all heuristics, therefore we base our PPO-BiHyb method on it.

**Learning Methods.** There are efforts to learn graph-wise similarity via deep learning [3, 34], which are regression models and ignores the combinatorial nature of graph similarity problems. [60] proposes neural-guided A* search, however, the learning is supervised. In this paper, we compare with PPO-Single by reimplementing [36] for GED. The model details and RL algorithm are kept in line with our PPO-BiHyb for fair comparison. The beam search width of PPO-Single is set to 20 so that the inference time of PPO-Single is comparable with our hybrid method. We also compare with Random-BiHyb which performs equal step numbers of random search w.r.t. PPO-BiHyb.

#### 4.2.2 Experimental Results

Results on GED are reported on AIDS dataset[4] containing chemical compounds for anti-HIV research [43]. The atoms are treated as nodes, and the atom types are encoded by one-hot features and we define an edit cost of 1 for different node types. Besides, we also define the cost for node/edge addition/deletion as 1. The AIDS dataset is split into three subsets w.r.t. the size of graphs, namely AIDS-20/30, AIDS-30/50, and AIDS-50+, and we exclude graphs smaller than 20 nodes because they are less challenging and can be solved exactly within several hours. We randomly build 50 training and 10 testing samples for all tests with fixed random seed.

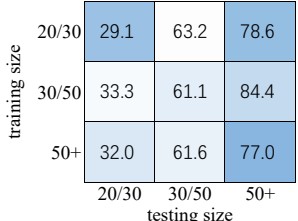

Figure 4: Sensitivity by training and testing sizes on AIDS.

The evaluation for GED on AIDS is presented in Tab. 2, where our method surpasses all learning-free heuristics and also performs better than single-level RL baseline PPO-Single. We also conduct a

---

[4]`https://wiki.nci.nih.gov/display/NCIDTPdata/AIDS+Antiviral+Screen+Data`

Table 2: Results on graph edit distance (GED) problems from AIDS dataset with the average number of nodes reported in brackets. The naming convention here AIDS-X/Y means the number of nodes are within the range of X and Y. PPO-Single can be viewed as our implementation of the peer RL method [36]. Here "objective" means the objective score i.e. the solved edit distance between two graphs, and "relative" is computed by the solved edit distance w.r.t. the best heuristic IPFP.

| dataset / method | AIDS-20/30 (#nodes=22.6) objective ↓ | relative ↓ | AIDS-30/50 (#nodes=37.9) objective ↓ | relative ↓ | AIDS-50+ (#nodes=59.6) objective ↓ | relative ↓ |
|---|---|---|---|---|---|---|
| Hungarian [44] | 72.9±19.2 | 94.9% | 153.4±28.0 | 117.9% | 225.6±33.9 | 121.4% |
| RRWM [9] | 72.1±23.7 | 92.8% | 139.8±31.9 | 98.6% | 214.6±41.3 | 110.6% |
| Hungarian-Search [45] | 44.6±8.5 | 19.3% | 103.9±22.7 | 47.6% | 143.8±31.5 | 41.1% |
| **IPFP** [6] | 37.4±8.5 | 0.0% | 70.4±15.1 | 0.0% | 101.9±13.1 | 0.0% |
| PPO-Single [36] | 56.5±14.4 | 51.1% | 110.0±19.2 | 56.3% | 183.9±16.9 | 80.5% |
| Random-BiHyb | 33.1±9.0 | -11.5% | 66.0±15.2 | -6.3% | 82.4±20.3 | -19.1% |
| **PPO-BiHyb (ours)** | **29.1±8.9** | **-22.2%** | **61.1±14.2** | **-13.2%** | **77.0±19.4** | **-24.4%** |

generalization study among different problem sizes, and Fig. 4 shows the objective scores from the sensitivity test for training/testing on GED problems with different sizes. The color map represents the percentage of improvement w.r.t. IPFP, where the darker color means more improvement. Our model can generalize to problem sizes unseen during training, but the generalized performance is usually inferior compared to training and testing with the same problem size.

### 4.3 Case 3: Hamiltonian Cycle Problem

The Hamiltonian cycle problem (HCP) arises from the notable problem of seven bridges of Königsberg proposed by Leonhard Eule [12]. Given a graph, the HCP arises as a decision problem on whether there exists a Hamiltonian cycle, which is known to be NP-complete [17]. We handle HCP by transforming it into the more general traveling salesman problem (TSP) in a fully-connected graph, where the existing edges in HCP are defined with length 0, and non-existing edges are with length 1. If a tour is found whose length is 0, then it is a Hamiltonian cycle. It is worth noting that HCP has not been discussed by most ML-TSP works [15, 28, 33, 57], which focus on the special case of Euclidean TSP with 2D coordinates.

#### 4.3.1 Implementation Components

**MDP Formulation.** The HCP instances are converted to TSP as discussed above, to leverage existing powerful TSP heuristics. As shown in Fig. 2(c), *state* is defined as the current graph $\mathcal{G}^k$ and *action* is defined as increasing an edge length in $\mathcal{G}^k$, resulting in a new graph $\mathcal{G}^{k+1}$. The increase of edge length softly prohibits the heuristic from traveling this edge and the LKH method [23] is adopted on graph $\mathcal{G}^{k+1}$ to obtain the new tour $\mathbf{x}^{k+1}$. The *reward* is the decrease of the current tour length w.r.t. the previous tour length: $f(\mathbf{x}^k|\mathcal{G}) - f(\mathbf{x}^{k+1}|\mathcal{G})$.

**State Encoder.** We encode HCP graph with GCN [32]. The node embeddings from GCN modules are then processed by an attention pooling layer to extract a graph-level embedding.

$$\mathbf{n} = \text{GCN}(\mathcal{G}^k), \ \mathbf{g} = \text{Att}(\mathbf{n}) \tag{10}$$

**Actor Net.** After obtaining the tour solved by the heuristic algorithm, we increase the length of any selected edge by 10 from the tour (10 is randomly set, and our approach seems not sensitive to this number), which empirically makes the edge harder to be selected in a tour later. The first action probabilities of selecting the starting node are predicted by a 3-layer ResNet block [22]. The second action is selecting the ending node, which is adjacent to the starting node on the tour and is predicted by an attention query.

$$P(\mathbf{a}_1) = \text{softmax}(\text{ResNet}_1([\mathbf{n} \, || \, \mathbf{g}])), \ P(\mathbf{a}_2|\mathbf{a}_1) = \text{softmax}\left(\mathbf{n} \cdot \text{tanh}(\text{Linear}(\mathbf{n}[\mathbf{a}_1]))^\top\right) \tag{11}$$

For training, we sample $\mathbf{a}_1, \mathbf{a}_2$ according to $P(\mathbf{a}_1), P(\mathbf{a}_2|\mathbf{a}_1)$. For evaluation, we perform a beam search with a width of 12 and maintain the top-12 actions by the improvement of the LKH algorithm.

**Critic Net.** It is built by max-pooling from all node features, and the pooled feature is concatenated with the graph-level feature from the state encoder, which are finally processed by a 3-layer ResNet.

$$\widetilde{V}(\mathcal{G}^k) = \text{ResNet}_2\left([\text{maxpool}(\mathbf{n}) \, || \, \mathbf{g}]\right) \tag{12}$$

**Heuristic Methods.** The performance of heuristics varies with the sizes and characteristics of instances. We choose three algorithms, *Nearest Neighbour* [42] who greedily travels to the next nearest node, *Farthest Insertion* [46] who repeatedly insert the non-traveled node with the farthest distance to the existing tour, and the third version of *Lin-Kernighan Heuristic (LKH3)* [23] which succeeds in discovering the best-known solutions for many TSP instances. For LKH3, we can set the number of random restarts (5 by default) to trade-off time for accuracy. We name the default LKH config as *LKH3-fast* which is also adopted for lower-level optimization in our PPO-BiHyb for its time-efficiency, and we also compare with *LKH3-accu* with 100 random restarts.

**Learning Methods.** Most deep learning TSP models are designated to handle fully connected 2D Euclidean TSP [15, 28, 33, 57], which is a different setting compared to our HCP testbed. Therefore, we implement PPO-Single following the most general framework [30], and the model details are kept in line with our PPO-BiHyb. We also compare with the Random-BiHyb baseline.

Table 3: Tests on FHCP with mean number of nodes in brackets. FHCP-X/Y means the number of nodes is in the range of X and Y.

| dataset method | FHCP-500/600 (#nodes=535.1) | |
|---|---|---|
| | TSP objective ↓ | found cycles ↑ |
| Nearest Neighbor [42] | 79.6±13.4 | 0% |
| Farthest Insertion [46] | 133.0±31.7 | 0% |
| **LKH3-fast** [23] | 13.8±25.2 | 0% |
| LKH3-accu [23] | **6.3±13.0** | 20% |
| PPO-Single [30] | 9.5±45.6 | 0% |
| Random-BiHyb | 10.0±21.9 | 0% |
| **PPO-BiHyb (ours)** | 6.7±14.0 | **25%** |

### 4.3.2 Experimental Results

We use the FHCP benchmark [21] composed of 1001 hard HCP instances. All instances are known to have valid Hamiltonian cycles, however, finding them is non-trivial for standard HCP/TSP heuristics. Table 3 shows the result from a subset of FHCP benchmark. As mentioned in Sec. 4.3.1, we convert HCP instances to binary TSP instances, whose tour lengths are denoted as "TSP objective". When the TSP objective is 0, it means that a Hamiltonian cycle is found. We use 50 training instances and 20 testing instances, and the model is trained on graphs of sizes from 250 to 500. Our method is comparative with the novel heuristic *LKH3-accu* and can even surpass in terms of found Hamiltonian cycles, which is the objective of HCP. We set the beamwidth of PPO-Single as 12 allowing its inference time to be $10\times$ longer than ours, but its performance is still inferior to our bi-level PPO-BiHyb. The Random-BiHyb baseline also improves the performance of LKH3-fast which is adopted for solving the lower-level problem.

## 5 Discussions

**Limitations in model design.** In this paper, we adopt the vanilla GCN implemented by TorchGeometric [13], and the detailed configurations can be found in Appendix C. The main purpose of our model design is to validate the effectiveness of the proposed bi-level optimization framework, and we adopt standard building blocks without heavy engineering, from which perspective we agree that there are room for further improvement. One possible direction may be adopting GNN-neural architecture search methods [16, 65].

**Potential negative impacts.** Our approach may potentially decrease the job opportunities and burden the workload of employees in companies, and it calls for the companies and social groups to take more responsible roles when facing the negative effects that come along with optimization tools.

**Conclusion.** We present a bi-level optimization framework based on hybrid machine learning and traditional heuristics, where the graph structure is optimized by RL to narrow down the feasible space of combinatorial problems. The new optimization problems are solved by fast heuristics. Experiments on large real-world combinatorial problems show the effectiveness of our approach. Our method also shows good generalization ability from small training instances to larger testing instances.

## Acknowledgments and Disclosure of Funding

This work was partly supported by National Key Research and Development Program of China (2020AAA0107600), Shanghai Municipal Science and Technology Major Project (2021SHZDZX0102), NSFC (U19B2035, 61972250, 72061127003), and Ant Group through Ant Research Program. The author Runzhong Wang was also partly supported by Wen-Tsun Wu Honorary Doctoral Scholarship, AI Institute, Shanghai Jiao Tong University. We would also like to thank Chang Liu, Jia Yan and Runsheng Gan for their valuable discussions when we were working on this paper.

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
