# A    Comparison with Other General MLCO Frameworks

Since obtaining ground-truth labels is non-trivial for NP-hard combinatorial tasks, there exist several efforts developing general MLCO methods without any requirement of ground-truth labels, including [8, 29, 30], our single-level baseline PPO-Single and our proposed PPO-BiHyb. Here we make a comparison concerning the model details and the capable problems of these methods.

Table 4: Comparison of general MLCO frameworks concerning their model details (upper half) and the capabilities to some popular CO problems according to our understanding (lower half).

| model | S2V-DQN [30] | NeuRewritter [8] | Erdős GNN [29] | PPO-Single | PPO-BiHyb (ours) |
|---|---|---|---|---|---|
| data type | graph | graph & sequence | graph | graph | graph |
| CO formulation | single-level | single-level | single-level | single-level | bi-level |
| learning method | RL (DQN) | RL (DQN) | unsup. learning | RL (PPO) | RL (PPO) |
| encoder | GNN | GNN or LSTM | GNN | GNN | GNN |
| decode strategy | greedy | local search | random sampling | beam search | beam search |
| graph cut | ✓ | ✓ | ✓ | ✓ | ✓ |
| vertex cover | ✓ | ✓ | ✓ | ✓ | ✓ |
| max clique | ✓ | ✓ | ✓ | ✓ | ✓ |
| DAG scheduling | ✓ | ✓ | × | ✓ | ✓ |
| graph edit distance | ✓ | ✓ | × | ✓ | ✓ |
| Hamiltonian cycle | ✓ | ✓ | × | ✓ | ✓ |
| traveling salesman | ✓ | ✓ | × | ✓ | ✓ |
| vehicle routing | ✓ | ✓ | × | ✓ | ✓ |
| expression simplify | × | ✓ | × | × | × |

We would also like to discuss the limitations of the approaches including ours. For S2V-DQN [30] and NeuRewritter [8], training the RL model is challenging due to the sparse reward and large action space issues especially for large-scale problems. Specifically, for graphs with $m$ nodes, the action space of S2V-DQN and NeuRewritter is $m$, and S2V-DQN requires $O(m)$ actions to terminate for most problems when the number of decisions is proportional to $m$. This conclusion also holds for PPO-Single. NeuRewritter may take any number of actions, but the more actions it takes, the more likely it will reach a better solution. In comparison, the action space of our PPO-BiHyb is $m^2$ (and decomposed as two sub-actions with space $m$), and the number of actions is restricted under 20 for graphs up to 2800 nodes. Therefore, from the perspective of learning, we cut the number of actions (i.e. mitigate the sparse reward issue) to ease RL training.

As shown in Tab. 4, the PPO-Single that serves as a baseline in our paper is designed following S2V-DQN [30], and we set the RL method as PPO and use beam search when decoding, to make a fair comparison between PPO-Single and PPO-ByHib.

Erdős GNN [29] is a novel framework with unsupervised learning, however, its main limitation is that this framework is incapable of handling constraints beyond simple node constraints. As shown in Tab. 4, NerRewritter is most general because it can be viewed as a learning-based local search meta-heuristic that represents the family of meta-heuristics including popular algorithms e.g. simulated annealing and genetic algorithm, and S2V-DQN and our PPO-BiHyb can handle most CO over graphs. It is also worth noting that there are some problems that are beyond our knowledge to tackle, e.g. the expression simplify problem, and it may requires experts with specific domain knowledge to generalize our proposed framework to more combinatorial problems.

# B    Implementation Details on PPO-Single

We have discussed the model details of PPO-BiHyb in Sec. 4, and in this section, we discuss the model details of the single-level RL baseline PPO-Single.

## B.1    Case 1: DAG Scheduling

**MDP Formulation.** Following the implementation from [39], the job nodes are scheduled in sequential order with PPO-Single. Here *state* is the current DAG $\mathcal{G}^k$ with a timestamp, and some of the nodes are already scheduled by the current timestamp. To represent the current state of the problem, the finished nodes, running nodes and unscheduled nodes are marked by different node

attributes, so that the state information is fully encoded by the nodes and edges of $\mathcal{G}^k$. *Action* is defined as scheduling the next node to be executed, and we define a "wait" action to move timestamp until the next event when any node finishes. After a node finishes, it will free some resources, and sometimes add some available nodes to be scheduled. The *reward* is computed as the negative makespan time.

**State Encoder.** In line with PPO-BiHyb, we also adopt GCN [32] to encode the state represented by DAG. Considering the structure of DAG, we design two GCNs: the first GCN processes the original DAG, and the second GCN processes the DAG with all edges reversed. The node embeddings from two GCN modules are concatenated, and an attention pooling layer is adopted to extract a graph-level embedding.

$$\mathbf{n} = \left[\text{GCN}_1(\mathcal{G}^k) \;\|\; \text{GCN}_2(\text{reverse}(\mathcal{G}^k))\right], \; \mathbf{g} = \text{Att}(\mathbf{n}) \tag{13}$$

**Actor Net.** Following PPO-BiHyb, the action net is built with a 3-layer ResNets block [22]. Since we are adopting a single-level RL method, only one action is selected at each step:

$$P(\mathbf{a}) = \text{softmax}(\text{ResNet}_1([\mathbf{n} \;\|\; \mathbf{g}])) \tag{14}$$

For training, we sample $\mathbf{a}$ according to $P(\mathbf{a})$. For testing, beam search is performed with a width of 10: actions with top-10 highest probabilities are searched, and all searched actions are evaluated by their current makespan, and only the best-10 actions are maintained for the next search step.

**Critic Net.** Also in line with PPO-BiHyb, the critic net is built by max pooling over all node features, and the pooled feature is concatenated with the graph feature from state encoder, and finally processed by another ResNet for value prediction.

$$\widetilde{V}(\mathcal{G}^k) = \text{ResNet}_2\left([\text{maxpool}(\mathbf{n}) \;\|\; \mathbf{g}]\right) \tag{15}$$

## B.2 Case 2: Graph Edit Distance

**MDP Formulation.** Our embodiment of PPO-Single on GED follows [36]. Since there are two graphs, we discriminate the graphs by subscripts $\mathcal{G}_1, \mathcal{G}_2$. The *state* is defined as the current partial solution between two graphs. At each *action*, the RL agent is required to firstly select a node from $\mathcal{G}_1$ and then select a node from $\mathcal{G}_2$. We further add "void nodes" to stand for node addition/deletion operations. *Reward* is defined as the negative edit length obtained by editing $\mathcal{G}_1$ to $\mathcal{G}_2$.

**State Encoder.** The state encoder is built with GCN [32] to aggregate graph features and the differentiable Sinkhorn-Knopp (SK) algorithm [49] for message passing across two graphs, and such designs are in line with our PPO-BiHyb.

$$\mathbf{n}_1 = \text{GCN}(\mathcal{G}_1), \; \mathbf{n}_2 = \text{GCN}(\mathcal{G}_2), \; \mathbf{S} = \text{SK}(\mathbf{n}_1 \mathbf{n}_2^\top) \tag{16}$$

The predicted doubly-stochastic matrix by SK is processed by considering the partial matching matrix. For PPO-Single, if one node is already matched, we assign the corresponding row/column in $\mathbf{S}$ as the known matching information, and we get the modified matrix $\mathbf{S}'$. The difference of node features between two graphs is obtained:

$$\mathbf{n} = \mathbf{n}_1 - \mathbf{S}' \cdot \mathbf{n}_2 \tag{17}$$

Graph-level features are obtained via attention pooling, which are fed to the critic net. It is worth noting that the matched nodes and the corresponding edges are occluded when computing the graph-level features.

$$\mathbf{g}_1 = \text{Att}(\mathbf{n}_1), \; \mathbf{g}_2 = \text{Att}(\mathbf{n}_2) \tag{18}$$

**Actor Net.** Since we are aiming to predict the edit operations from $\mathcal{G}_1$ to $\mathcal{G}_2$, the actor net is designed to selected two nodes from two graphs separately. The node selection from the first graph is done by a ResNet module, and the node selected from the second graph is done by an attention query based on the selected first node.

$$P(\mathbf{a}_1) = \text{softmax}\left(\text{ResNet}(\mathbf{n})\right), \; P(\mathbf{a}_2|\mathbf{a}_1) = \text{softmax}(\mathbf{n}_2 \cdot \tanh(\text{Linear}\left(\mathbf{n}[\mathbf{a}_1]\right))^\top) \tag{19}$$

$P(\mathbf{a}_1), P(\mathbf{a}_2|\mathbf{a}_1)$ are used to sample the actions $\mathbf{a}_1, \mathbf{a}_2$ in training, and the beamwidth is set as 10 for evaluation.

**Critic Net.** Following PPO-BiHyb, the graph-level features are processed by the neural tensor network (NTN) [50] followed by 2 fully-connected regression layers whose output is the one-dimensional value prediction:

$$\widetilde{V}(\mathcal{G}_1, \mathcal{G}_2) = \text{fc}\left(\text{NTN}\left(\mathbf{g}_1, \mathbf{g}_2\right)\right) \tag{20}$$

### B.3 Case 3: Hamiltonian Cycle Problem

**MDP Formulation.** Since we handle HCP by transforming it to a TSP, our implementation of PPO-Single on HCP is in line with [30], where *state* is defined as the graph $\mathcal{G}^k$, and the current partial tour is encoded by the edges of $\mathcal{G}^k$ as a weight of 10. In comparison, the edge weights of the original graph can be only 0 or 1. *Action* is defined as selecting the next node to be visited. The *reward* is defined as the negative value of the current tour length.

**State Encoder.** Following PPO-BiHyb, we encode HCP graph with GCN [32]. The node embeddings from GCN modules are then processed by an attention pooling layer to extract graph-level embeddings.

$$\mathbf{n} = \text{GCN}(\mathcal{G}^k), \ \mathbf{g} = \text{Att}(\mathbf{n}) \tag{21}$$

**Actor Net.** Following [30, 57], the action of selecting the next visited node is achieved by attention. More specifically, the attention query is the feature of the previous visited node, and the keys are the features of all non-visited nodes.

$$P(\mathbf{a}^k|\mathbf{a}^{k-1}) = \text{softmax}\left(\mathbf{n} \cdot \tanh(\text{Linear}(\mathbf{n}[\mathbf{a}^{k-1}]))^\top\right) \tag{22}$$

Here $\mathbf{a}^k$ means the action at step $k$, and $\mathbf{a}^0$ is fixed as the first node. For training, we sample $\mathbf{a}^k$ according to $P(\mathbf{a}^k|\mathbf{a}^{k-1})$. For evaluation, we perform a beam search with a width of 12 and maintain the best-12 actions by the tour length.

**Critic Net.** In line with PPO-BiHyb, the critic net is built by max-pooling from all node features, and the pooled feature is concatenated with the graph-level feature from the state encoder. The features are then processed by a 3-layer ResNet module.

$$\widetilde{V}(\mathcal{G}^k) = \text{ResNet}\left([\text{maxpool}(\mathbf{n}) \ || \ \mathbf{g}]\right) \tag{23}$$

## C Model Hyperparameters

In this section we describe the model parameters for both PPO-Single in Tab. 5 and PPO-BiHyb in Tab. 6. The model structures are kept to be the same, and some training/evaluation statistics might be different concerning the differences between single-level and bi-level frameworks. Take the DAG scheduling problem as an example, the number of actions in each episode is different between these two methods. For a scheduling problem with $m$ nodes, the single-level PPO-Single requires more than $m$ actions to reach a complete solution (including the "wait" actions we defined for PPO-Single), but the number of actions in PPO-BiHyb is restricted to be $K$, and we empirically set $K = 20$ to produce satisfying results. Since the PPO-Single is with a longer episode, we set larger $\gamma = 0.99$ hoping the agent will focus more on the long-term reward in the long episode. We also increase the number of timesteps between two model updates for PPO-Single to reduce the variance of gradients. The width of beam search is also different to ensure that the inference time of PPO-Single is no less than ours, but as shown in the main paper, the performance of PPO-Single is still inferior to our PPO-BiHyb given more time budgets. We set the upper-limit of training time to be 48 hours for all models.

We also set separate learning rates for the GNN module and the other modules, because we empirically find the GNN easy to collapse when learning with reinforcement learning on combinatorial

Table 5: Hyperparameter configs for PPO-BiHyb on DAG scheduling, GED and HCP problems.

| | DAG scheduling | GED | HCP | description |
|---|---|---|---|---|
| heuristic name | Critical Path | IPFP | LKH-fast | heuristic algorithm for lower-level optimization |
| $\gamma$ | 0.95 | 0.95 | 0.95 | discount ratio for accumulated reward |
| $\epsilon$ | 0.1 | 0.1 | 0.1 | controls the trust region of PPO |
| $K$ (#actions) | 20 | 10 | 8 | max number of modified edges (number of actions) |
| #epoches | 10 | 10 | 10 | number of gradient updates at each update |
| update timestep | 20 | 10 | 8 | number of timesteps between two updates |
| GNN learning rate | $1 \times 10^{-4}$ | $1 \times 10^{-4}$ | $1 \times 10^{-4}$ | learning rate for the state encoder GNN |
| learning rate | $1 \times 10^{-3}$ | $1 \times 10^{-3}$ | $1 \times 10^{-3}$ | learning rate for other modules |
| #GNN layers | 5 | 3 | 3 | number of GNN layers |
| node feature dim | 64 | 64 | 16 | dimension of the output node feature of GNN |
| beamwidth | 3 | 3 | 12 | beam search width |

Table 6: Hyperparameter configs for PPO-Single on DAG scheduling, GED and HCP problems.

| | DAG scheduling | GED | HCP | description |
|---|---|---|---|---|
| $\gamma$ | 0.99 | 0.99 | 0.99 | discount ratio for accumulated reward |
| $\epsilon$ | 0.1 | 0.1 | 0.1 | controls the trust region of PPO |
| #actions | 5000 | 200 | 600 | max number of actions |
| #epoches | 10 | 10 | 10 | number of gradient updates at each update |
| update timestep | 50 | 50 | 20 | number of timesteps between two updates |
| GNN learning rate | $1 \times 10^{-4}$ | $1 \times 10^{-4}$ | $1 \times 10^{-4}$ | learning rate for the state encoder GNN |
| learning rate | $1 \times 10^{-3}$ | $1 \times 10^{-3}$ | $1 \times 10^{-3}$ | learning rate for other modules |
| #GNN layers | 5 | 3 | 3 | number of GNN layers |
| node feature dim | 64 | 64 | 16 | dimension of the output node feature of GNN |
| beamwidth | 10 | 10 | 12 | beam search width |

optimization. Besides, there are some recent attempts (e.g. submission in this link[5]) studying the power of randomly initialized GNNs. We empirically find the randomly initialized GNN learned with reduced learning rate produces satisfying result with our MLCO testbed.

# D    Inference Time

In this section, we report the inference time of both PPO-Single baseline and our PPO-BiHyb on all test cases, as shown in Tab. 7.

Table 7: Comparison of inference time (in seconds) of PPO-Single [39, 36, 30] and PPO-BiHyb (ours) on all problems considered in this paper. We control the beam search size of PPO-Single to ensure that its inference time is no less than ours.

| dataset | TPC-H (DAG scheduling) | | | AIDS (GED) | | | FHCP (HCP) |
|---|---|---|---|---|---|---|---|
| problem size | 50 | 100 | 150 | 20/30 | 30/50 | 50+ | 500/600 |
| PPO-Single [39, 36, 30] | 415.4 | 2761.6 | 6852.3 | 288.4 | 982.0 | 1350.4 | 3322.4 |
| PPO-BiHyb (ours) | 48.7 | 110.1 | 407.9 | 224.6 | 513.2 | 1297.9 | 272.2 |

It is worth noting that our PPO-BiHyb is slower than the heuristic algorithm adopted for lower-level optimization because this heuristic algorithm is called multiple times in the RL routine. We also consider more time-consuming heuristics, and the following heuristics considered in our experiments are with a comparable inference time with PPO-BiHyb: Hungarian-search for GED and LKH-accu for HCP. For DAG scheduling, we tried by firstly formulating DAG scheduling as an integer linear programming, and then solving the relaxed linear programming problem by the GLOP solver delivered with Google ORTools. However, it did not converge within 24 hours for even the smallest test set (TPC-H-50), thus we gave up this baseline.

# E    A Study on How RL Modifies the Graph

In this section we include a brief study on the AIDS dataset for graph edit distance problem about the relation between the number of actions taken by RL and the reward. Here we list the average number of actions, the average reward values, and the Pearson correlation for all test instances:

Table 8: For PPO-BiHyb on AIDS dataset, the average number of actions and the reward, and the Pearson correlation between number of actions and the reward.

| AIDS-20/30 | | AIDS-30/50 | | AIDS-50+ | |
|---|---|---|---|---|---|
| # act | reward | # act | reward | # act | reward |
| 4.1 | 8.3 | 2.7 | 9.3 | 3.4 | 24.9 |
| $\rho = 0.054$ | | $\rho = 0.075$ | | $\rho = 0.680$ | |

[5]https://openreview.net/forum?id=L7Irrt5sMQa

The average number of actions is relatively small with respect to the number of nodes, suggesting that in general, the RL agent learns to modify a small fraction of "critical edges" that can lead to a large reward. Besides, we also compute the Pearson correlation between the number of actions and the reward, and only the largest AIDS-50+ has a large correlation. It is not surprising because a larger graph should require more modifications to achieve a better reward, and for smaller graphs, such correlation seems not significant.

## F    Computer Resources

Experiments on DAG scheduling and GED are conducted on an internal cluster with Tesla P100 GPU, Intel E5-2682v4 CPU @ 2.50GHz and 128GB memory. Experiments on HCP are conducted on our workstation with RTX2080Ti GPU, Intel i7-7820X CPU @ 3.60GHz and 64GB memory. All experiments are run with only one GPU.

## G    Broader Impact

In this paper, we propose a general ML approach to solve CO over graphs, whose applications can be found in scheduling real-world tasks to increase efficiency. We would like to discuss the broader impact of this paper according to the following aspects:

*a) Who may benefit from this research.* The companies and organizations who use our optimization technique and their shareholders may benefit from this research, as our technique is aimed to increase efficiency, cut expenses and increase profits. Besides, a broader range of people may also benefit, because an increased efficiency usually means a save of resources and perhaps less pollution, which will further benefit the whole society.

*b) Who may be put at risk from this research.* The proposed optimization method may take the place of some job positions e.g. dispatchers who used to perform the optimization tasks with human expertise. The optimization method may also burden the workload of employees when pursuing the optimization goal. Therefore, it might be appealing to take account of the happiness of employees when scheduling, and it calls for more responsibilities of the company to provide training programs and new positions for the affected people and to care more about the rights of the workers.

*c) What are the consequences of failure of the system.* A failure of our ML-based CO approach will fail to cut the cost of finishing the task. More specifically, the performance will degenerate to be the same as the performance of the heuristic method adopted to solve the lower-level optimization. In most cases, the underlying task will not fail because the heuristic will still provide a feasible (although sub-optimal) solution.

## H    Licenses and Further Information on Used Assets

The following datasets and codebases are used for this research and we list their license information as follows:

- The usage of TPC-H dataset is under clause 9 of the End-User License Agreement (EULA) of TPC[6]: "THE TPC SOFTWARE IS AVAILABLE WITHOUT CHARGE FROM TPC".
- The AIDS dataset is collected by [43] and is free for academic use.
- The open-source repository GEDLIB [5] is under LGPL-3.0 License.
- The FHCP challenge dataset is publicly available and the authors require to cite their paper [21] in new publications.
- LKH-3 is an implementation of the Lin-Kernighan traveling salesman heuristic, which is described in [24]. The code is distributed for research use. The author reserves all rights to the code.

All datasets and codebases are publicly available. The datasets cover problems that arise from business computing, anti-HIV molecules and pure Hamiltonian graph data, which are not closely related to human identities and shall not contain offensive or biased information.

---

[6]http://tpc.org/TPC_Documents_Current_Versions/txt/TPC-EULA_v2.2.0.txt