# OpenReview forum: "A Bi-Level Framework for Learning to Solve Combinatorial Optimization on Graphs"
_NeurIPS.cc/2021/Conference — NeurIPS 2021 Poster_

### Official Review · Reviewer_ovsi · 2021-07-15

**Rating:** 7
**Confidence:** 4

**Summary:**

This paper is part of the newly emerging literature on solving combinatorial optimization problems using machine learning approaches (particularly reinforcement learning). It focuses on problems on graphs. It proposes a new hybrid approach which uses RL to modify the structure of the input graph into a surrogate graph, which is then given to an existing heuristic. The idea is that this simplifies the task for RL (since it only needs to modify the original graph instead of solving the problem) while allowing us to improve the performance of the heuristic. Experiments are given on three example tasks, showing that the proposed method improves over the heuristics on their own in terms of solution quality, and also outperforms an attempt to solve the entire task with RL.


**Limitations And Societal Impact:**

yes

**Main Review:**

To my knowledge, the idea of using RL to modify the input graph is novel. It represents a nice way to combine data-driven and algorithmic means of solving optimization problems. The experimental results show an improvement over the individual heuristics used, indicating that learning to modify the structure of the graph can actually improve the performance of the heuristic. Accordingly, I think that the paper does make a valuable contribution.

A few comments/questions:

First, test sets are often small (10-20 instances). Error bars should be reported to see if the improvements are significant (and/or more instances should be added to the test set). This is important to assess the validity of the results, aside from other comments about adding new content to the paper.

Second, it would be preferable to show results on at least one domain for which an existing RL method can be applied exactly. Developing and tuning a RL solver for an optimization problem can require a lot of effort, so to provide the most informative comparison between the single and bi-level frameworks, it would be good to see an example where another set of authors has produced a strong baseline.

Third, how does the inference time compare to the runtime of the heuristics? I.e., does RL add significantly to the runtime cost?

Fourth, is it possible to develop an understanding of how RL tends to modify the input graph?


**Time Spent Reviewing:**

1

---

> ### Author Response · Authors · 2021-08-09
> **Response to Reviewer ovsi**
>
> Your precious comments offer us a lot to explore, which we deeply cherish. We set out below our responses to each of the questions.
>
>
>
> **Q1: Report error bars**
>
> * RE: Many thanks for your suggestions. Here we list the variance values on objectives of PPO-BiHyb and heuristic algorithms as follows.
>
>
>
> **DAG scheduling**
>
>   |                    | TPC-H-50 (#nodes=467.2) |  | TPC-H-100 (#nodes=929.8) |  | TPC-H-150 (#nodes=1384.5) |            |
>   | ------------------ | ----------------------- | ------------------------ | ------------------------- | ---------- | ------------------ | --------- |
>   |                    | objective $\downarrow$               | relative $\downarrow$                 | objective $\downarrow$                 | relative $\downarrow$   | objective $\downarrow$          | relative $\downarrow$  |
>   | shortest job first | 12818$\pm$2214          | 30.5%                    | 19503$\pm$3260            | 15.3%      | 27409$\pm$2748     | 12.2%     |
>   | tetris scheduling  | 12113$\pm$1398          | 23.3%                    | 18291$\pm$2223            | 8.1%       | 25325$\pm$2842     | 3.7%      |
>   | critical path      | 9821$\pm$1176           | 0.0%                     | 16914$\pm$2499            | 0.0%       | 24429$\pm$2484     | 0.0%      |
>   | PPO-BiHyb (ours)   | **8906$\pm$922**        | **-9.3%**                | **15193$\pm$2275**        | **-10.2%** | **22371$\pm$2538** | **-8.4%** |
>
> ​  **Graph edit distance**
>
>   |                  | AIDS-20/30 (#nodes=22.6) |  | AIDS-30/50 (#nodes=37.9) |  | AIDS-50+ (#nodes=59.6) |            |
>   | ---------------- | ------------------------ | ------------------------ | ---------------------- | ---------- | ----------------- | ---------- |
>   |                  | objective $\downarrow$                | relative $\downarrow$                 | objective $\downarrow$              | relative $\downarrow$   | objective $\downarrow$         | relative $\downarrow$   |
>   | Hungarian        | 72.9$\pm$19.2            | 94.9%                    | 153.4$\pm$28.0         | 117.9%     | 225.6$\pm$33.9    | 121.4%     |
>   | RRWM             | 72.1$\pm$23.7            | 92.8%                    | 139.8$\pm$31.9         | 98.6%      | 214.6$\pm$41.3    | 110.6%     |
>   | Hungarian-Search | 44.6$\pm$8.5             | 19.3%                    | 103.9$\pm$22.7         | 47.6%      | 143.8$\pm$        | 41.1%      |
>   | IPFP             | 37.4$\pm$8.5             | 0.0%                     | 70.4$\pm$15.1          | 0.0%       | 101.9$\pm$13.1    | 0.0%       |
>   | PPO-BiHyb (ours) | **29.1$\pm$8.9**         | **-22.2%**               | **61.1$\pm$14.2**      | **-13.2%** | **77.0$\pm$19.4** | **-24.4%** |
>
>   **Hamiltonian cycle problem**
>
>   |                    | FHCP-500/600 (#nodes=535.1) |              |
>   | ------------------ | --------------------------- | ------------ |
>   |                    | TSP objective $\downarrow$               | found cycles $\uparrow$ |
>   | Nearest Neighbor   | 79.6$\pm$13.4               | 0%           |
>   | Farthest Insertion | 133.0$\pm$31.7              | 0%           |
>   | LKH3-fast          | 13.8$\pm$25.2               | 0%           |
>   | LKH3-accu          | **6.3$\pm$13.0**                | 20%          |
>   | PPO-BiHyb (ours)   | 8.5$\pm$18.7                | **25%**          |
>
>   We will update the error bars in the final version.
>
>
>
> **Q2: Experiment in domains with existing RL methods**
>
> * RE: Thank you for the valuable comment, but we might not be able to provide new results because the workload of experimenting with an additional problem may exceed this rebuttal period: we need to develop the RL environment, learning-free baselines, and both PPO-Single and PPO-BiHyb need to be carefully tuned.
>
>   The problems considered in this paper are seldom considered by the ML community, compared to trending topics like TSP. The aim of this paper is also to inspire more researchers to exploit their domain knowledge and develop various ML approaches for different downstream tasks beyond problems such as TSP. It may also help to influence a wider range of audiences with the recently developed MLCO approaches from the ML community.
>
>   In future work, we plan to include experiments in domains with existing RL methods to validate the effectiveness of this proposed approach, and to better position this paper with previous works.
>
>
>
> **Q3: About the runtime of RL method and heuristics**
>
> * RE: The RL method is slower than the heuristic algorithm adopted for lower-level optimization because this heuristic algorithm is called multiple times in the RL routine. In the experiment, we also compare heuristics with a comparable time cost with PPO-BiHyb: Hungarian-search for GED and LKH-accu for HCP. For DAG scheduling, we tried by firstly formulating DAG scheduling as an integer linear programming, and then solving the relaxed linear programming problem with Google ORTools. However, it did not converge within 24 hours for even the smallest test set (TPC-H-50), thus we gave up this baseline.
>
>
>
> **Q4: Understanding how RL modifies the graph**
>
> * RE: Many thanks for your suggestion. We analyze the AIDS dataset for graph edit distance problem about the relation between the number of actions taken by RL and the reward.
>
>   Here we list the average number of actions, the average reward values, and the Pearson correlation for all test instances:
>
>   | AIDS-20/30 |        | AIDS-30/50 |        | AIDS-50+ |        |
>   | ---------- | ------ | ---------- | ------ | -------- | ------ |
>   | nact       | reward | nact       | reward | nact     | reward |
>   | 4.1        | 8.3    | 2.7        | 9.3    | 3.4      | 24.9   |
>   | $\rho=$    | 0.054  | $\rho=$    | 0.075  | $\rho=$  | 0.680  |
>
>   The average number of actions is relatively small with respect to the number of nodes, suggesting that in general, the RL agent learns to modify a small fraction of "critical edges" that can lead to a large reward. Besides, we also compute the Pearson correlation between the number of actions and the reward, and only the largest AIDS-50+ has a large correlation. It is not surprising because a larger graph should require more modifications to achieve a better reward, and for smaller graphs, such correlation seems not significant.
>
>
>
> We hope the above elaborations could help you get through the confusion. We remain available for any further questions you may have and we look forward to your feedback at your earliest convenience.

---

### Official Review · Reviewer_RkUB · 2021-07-17

**Rating:** 4
**Confidence:** 4

**Summary:**

This paper proposes a hybrid bi-level method for combinational optimization problems on graphs, which combines traditional heuristics for combinational optimization and machine learning including reinforcement learning, graph neural networks, etc.

**Limitations And Societal Impact:**

Please see "Cons" above.

**Main Review:**

Pro：
1. The problem is important and meaningful.
2. This paper proposes a bi-level hybrid solver for graphs, which can be used for several tasks on graphs.
3. The authors conducted experiments on several different datasets for various tasks, which demonstrates that the model is general and effective.

Cons:
1. The codes are not submitted, which raises concerns about reproducing the results.
2. This paper overclaims on this point: “we take an initiative for developing a general hybrid approach that integrates traditional heuristic solvers with machine learning algorithm”
3. It is not clear about the range of the problems that the framework is useful for.

Details:

The idea of combining heuristics for combinational optimization and machine learning for solving combinational optimization problems on graphs is interesting. I agree that although deep learning is powerful, it is usually unfeasible to use deep learning to directly solve some hard combinational optimization problems on graphs. The experimental results show that this model has the advantages of two parts: automated editing the graph with reinforcement learning and efficiently solving the remaining problem by heuristics for combinational optimization.

However, I am confused about the following issues, and I hope the authors can answer these questions in the rebuttal stage,
1. It would be better if the authors can provide several examples for the proposition (line 119) for understanding. What is the exact meaning of “If all constraints are encoded by the graph structure”? Can it be formally defined? What problems can be solved and what cannot? The related statements are vague.

2. Furthermore, the proof states that given G and G’, we can find constraints that make x* is the output of a given heuristic algorithm. In my opinion, it’s different from given G and constraints, we can find G* that x* would be the solution (what the proposition proposes).

3. It would be better if the authors can provide more details about the training process and plot the learning curves which can help us understand how the convergence process is.

4. The authors should provide more examples in experiments as case studies.

5. The following is overclaimed, “we take an initiative for developing a general hybrid approach that integrates traditional heuristic solvers with machine learning algorithm”. Actually, there are existing works that combine these two things. For example, meta-learning for heuristic algorithm search can combine them. This area starts in 1970 and has many papers. For recent ones, please check Frank Hutters' papers which are published around 2010.

6. It seems RL is not the only method for searching for the graph, the authors should compare their RL-based method with evolutional search and stochastic search, etc., which can be used for the graph search/modification.

7. The motivation of some designs is not clear, e.g., the design of the GCN module in Sec. 4.1.1 (maybe because of the page limit). Because the problem is defined based on graphs, how to use or design GCN would be key, but the description in this paper about GCN design is very short.


**Time Spent Reviewing:**

5

---

> ### Author Response · Authors · 2021-08-09
> **Response to Reviewer RkUB**
>
> We appreciate the reviewer for the thought-provoking comments and thank you for identifying the novelty and strong experiment results of our paper. We also truly agree with the reviewer that “it is usually unfeasible to use deep learning to directly solve some hard combinatorial optimization problems on graphs”, and our approach “has the advantages of two parts: … reinforcement learning and … heuristics...” We elaborate the Proposition and add a random search baseline named Random-BiHyb. In the final version, we will better discuss related works and publish the code.
>
> We set out below our responses to each of the questions.
>
> **Q1: The Proposition is unclear and** **provide examples for the Proposition**
>
> * RE: **About the Proposition.** We regret that the Proposition may be unclear due to limited pages. To better clarify our point, we slightly reword the original Proposition as:
>
>   **Proposition.** _Suppose the optimal solution $\mathbf{x}^*$ to $\mathcal{G}$ is given, at least one instance of $\mathcal{G}^*$ can be constructed by modifying $\mathcal{G}$, such that $\mathcal{G}^*$ has the only feasible solution of $\mathbf{x}^*$._
>
>   _Proof._ The optimal solution $\mathbf{x}^*$ to $\mathcal{G}$ must satisfy all the constraints in the original graph, $\mathcal{G}$. If $\mathbf{x}^*$ is not the only feasible solution to $\mathcal{G}$, we can add constraints to force that $\mathbf{x}^*$ is the only solution, which is equal to modify $\mathcal{G}$ to construct $\mathcal{G}^*$, such that $\mathbf{x}^*$ is the only feasible solution to $\mathcal{G}^*$.
>
>   **Examples.** We take _DAG scheduling_ as an example to clarify this Proposition. Without loss of generality, suppose $\mathbf{x}^*$ be the sequence of nodes $1$ to $n$ in order. Correspondingly, we can modify the original graph $\mathcal{G}$ to acquire $\mathcal{G}^*$ to which $\mathbf{x}^*$ is the only feasible solution: if any edge connecting two adjacent nodes in the sequence, e.g. $(i, i+1)$ ($0 < i < n$), does not exist in $\mathcal{G}$, it is added to $\mathcal{G}$. As a result, this exact node sequence from $1$ to $n$, $\mathbf{x}^*$, is the only feasible solution, which will be also the output of any heuristic algorithm. Take another example, if two nodes $i$ and $i+1$ need to be executed in parallel, we can add two edges: $(i-1, i)$ and $(i-1, i+1)$.
>
>   **Other remarks.** About “...it is different from given G and constraints, we can find G* that x* would be the solution”:
>
>   This Proposition provides a theoretical grounding for developing graph-modification methods to tackle combinatorial problems. It is worth noting that $\mathcal{G}^{\*}$ only suggests that graph modification is a promising direction, and finding $\mathcal{G}^{\*}$ given $\mathcal{G}$ is nearly infeasible in practice. In this paper, we propose to improve the solving quality for heuristic algorithms by finding optimized (not necessarily optimal) graphs by learning based on the bi-level reformulation of the original single-level problem.
>
>     For more details, please refer to  the section **“We elaborate the Proposition”** in https://openreview.net/forum?id=nwWLJsTJfv&noteId=fWczDIedyzE
>
> **Q2: Compare with evolutionary search and stochastic search**
>
> * RE: We add a new entry named **Random-BiHyb** for all experiments. Random-BiHyb performs random search *instead of PPO* when modifying the graph, and its search steps are equal to PPO-BiHyb.
>
>   **Result analysis.** In summary, Random-BiHyb can improve the lower-level heuristic algorithm, suggesting the effectiveness of the bi-level optimization pipeline. Besides, Random-BiHyb is inferior to PPO-BiHyb, suggesting that PPO is more effective than naive random search.
>
>   Please refer to the Section **"We add a new entry Random-BiHyb"** https://openreview.net/forum?id=nwWLJsTJfv&noteId=fWczDIedyzE
>
>   **Evolutionary search and beyond.** Implementing and tuning evolutionary search methods may be too time-consuming for this short rebuttal period, and we are also aware that black-box optimization methods may be another interesting line of work. We plan to discuss and compare them in future work. Besides, even if the evolutionary search shows convincing results, it does not conflict with the main contribution of this paper: a bi-level optimization framework for combinatorial optimization on graphs. We are also aware that both evolutionary search and RL show positive results on many combinatorial problems, and we refer to RL in this paper for the purpose of building a learnable framework.
>
> **Q3: About the code**
>
> * RE: We regret to say that the code is currently under internal reviews before being made public. Once this paper is accepted, we will make every effort to publish the code, datasets, and pretrained models.
>
> **Q4:** **About previous ML-CO methods e.g. papers by Prof. Frank Hutters**
>
> * RE: Thank you for your suggestions. We will better position this paper with previous works in the final version.
>
> **Q5: About the range of the problems that this framework is useful for. What problems can be solved and what cannot?**
>
> * RE: If the Proposition can be achieved by modeling the original problem with suitable graph structures, it is a sufficient condition that this problem can be handled by our framework.
>
>   For a specific problem, modeling it by a reasonable graph structure is non-trivial and requires domain-specific knowledge. Fortunately, we manage to identify the following problems that can be modeled under our proposed framework, as summarized in Table 4 in the supplementary material:
>
>   * graph cut
>   * vertex covering
>   * max clique
>   * DAG scheduling
>   * graph edit distance
>   * Hamiltonian cycle
>   * traveling salesman
>   * vehicle routing
>
>   We also list the following problem that is beyond our knowledge, which may require domain-specific knowledge for problem modeling:
>
>   * expression simplify
>
>   Since there are so many combinatorial problems beyond the popular ones in the ML community like TSP, our wish for this paper is to inspire more researchers to leverage their expert knowledge and develop methods for various downstream combinatorial tasks. By writing this paper, we also wish to influence a wider scope of audiences with recently developed machine learning combinatorial optimization techniques.
>
> **Q6:** **More details about the training process and plot the learning curves**
>
> * RE: Thank you for the suggestions. We will include these materials in the appendix of the final version.
>
> **Q7: More case studies in experiments**
>
> * RE: The 3 cases studied in this paper belong to 3 important categories of real-world combinatorial problems: scheduling (DAG scheduling), matching (GED) and routing (HCP), which are seldom considered by previous machine learning papers compared to popular topics like TSP. The workload of adding an additional problem may exceed this rebuttal period: we need to develop the RL environment, learning-free baselines, and both PPO-Single and PPO-BiHyb need to be carefully tuned. Besides, in comparison, most existing machine learning combinatorial optimization papers study 2-4 cases which are in line with our paper:
>
>   * (3 cases) Dai et al. *Learning Combinatorial Optimization Algorithms over Graphs.* NeurIPS 2017.
>   * (4 cases) Li et al. *Combinatorial Optimization with Graph Convolutional Networks and Guided Tree Search*. NeurIPS 2018.
>   * (3 cases) Chen and Tian. *Learning to Perform Local Rewriting for Combinatorial Optimization*. NeurIPS 2019.
>   * (2 cases) Karalias and Loukas. *Erdo˝s Goes Neural: an Unsupervised Learning Framework for Combinatorial Optimization on Graphs*. NeurIPS 2020.
>
>   There are also papers that focus on a single case:
>
>   * (1 case) Mao et al. *Learning Scheduling Algorithms for Data Processing Clusters*. SIGCOMM 2019.
>   * (1 case) Yolcu and Poczos. *Learning to solve circuit-SAT: An unsupervised differentiable approach*. ICLR 2019.
>   * (1 case) Zhang et al. *Learning to Dispatch for Job Shop Scheduling via Deep Reinforcement Learning*. NeurIPS 2020.
>   * (1 case) Lu et al. *A Learning-based Iterative Method for Solving Vehicle Routing Problems.* ICLR 2020.
>
>   We would like to highlight that we include a Proposition as the theoretical grounding of our proposed framework. In contrast, these previous works are empirical and do not provide any theoretical analysis.
>
>   We plan to explore more challenging and important combinatorial problems and validate the effectiveness of our approach in feature work. By writing this paper, we are also looking forward to inspiring more researchers to adopt our framework to tackle various downstream tasks which are seldom considered by the ML community, hoping to influence a wider range of audiences of this paper and other advances from the MLCO community.
>
> **Q8: About the design of GCNs**
>
> * RE: We adopt the vanilla GCN implemented by [torch_geometric](https://pytorch-geometric.readthedocs.io/en/latest/modules/nn.html#torch_geometric.nn.conv.GCNConv). The detailed configurations can be found in Section C of the supplementary material. Our main purpose is to validate the effectiveness of the proposed bi-level optimization framework, and the motivation in our model design is to use standard building blocks without heavy engineering on the model structure.
>
>   According to the new entry Random-BiHyb (see Q2), even a vanilla random search can achieve significant improvement based on our bi-level framework. Since we mainly focus on the general framework, we leave room for further improvement from the perspective of model design. One possible direction may be adopting GNN-neural architecture search methods, e.g.
>
>   * Zhao et al. *Search to aggregate neighborhood for graph neural network.* ICDE 2021.
>   * Yang et al. *Graph Neural Architecture Search.* IJCAI 2020.
>
> We hope the above elaborations could help you get through the confusion. We remain available for any further questions you may have and we look forward to your feedback at your earliest convenience.

---

### Official Review · Reviewer_wSnT · 2021-07-17

**Rating:** 5
**Confidence:** 4

**Summary:**

While many approaches have been proposed for the combinatorial optimization on graphs, authors propose a bi-level framework that incorporates the conventional heuristic solvers into MDP setting. In particular, three kinds of combinatorial problems -- DAG scheduling, graph edit distance, and Hamiltonian cycle problem (HCP) -- are solved by the proposed framework. The main idea of the framework is to apply RL to modify a given (constraint) graph so that a heuristic algorithm over the modified graph can result in the (near-)optimal solution of the original problem. For the framework to work well, the modification of the graph should occur in the direction that 1) the feasible space needs to be reduced and 2) the updated objective function value should be minimized in the end. Experimental results show that the proposed method outperform baseline models on DAG scheduling and graph edit distance problem, while showing comparable performance with the best baseline model on HCP.

The main contributions of the paper is as follows. First, the idea of modifying original graph constraints to find better solutions by using heuristic algorithms is very unique and worth checking. Second, the combination of heuristic algorithms and RL is also a nice direction of solving the combinatorial problem. Finally, authors formulate the three kinds of typical combinatorial problems into the proposed framework and show experimental results on benchmark datasets.

**Limitations And Societal Impact:**

Authors provide enough limitations societal impact.

**Main Review:**

Originality
Combinatorial optimization on graphs is very broad and authors focus on conventional problems -- DAG scheduling, graph edit distance, and HCP. And the proposed framework to combine MDP and heuristic algorithms is novel because of apply the RL to modify graphs in a certain direction. This direction is very different from a lot of work using RL that directly solves a given problem.

Quality
While the proposed idea is interesting, some technical aspects are unclear or unconvincing. Four main unclear points are following.

1) Proposition in Page 3 does not seem proven. There is no guarantee that x* is the only feasible solution of G* when it is solved by a given heuristic algorithm. Without the proof of that unique solution guarantee, x* is not necessarily a solution of the heuristic solution. While it is not a trivial proof, authors skip this part and conclude.

2) Intuitively, adding more constraints is likely to make the problem harder even for a given heuristic algorithm. Hence, we can imagine that the heuristic algorithm is likely to find a worse solution when more constraints are added. Given the counterintuitive situation (at least apparently), the explanation why the proposed framework will result in a better solution with a fair probability is missing.

3) Experimental setting is not very convincing. Since heuristic algorithms result in local optimal solutions typically depending on initial settings, more trials with memorization of best results are likely to produce better performance. For a fair comparison, the proposed framework should be also compared with any heuristic MDP replacement such as identity (just more trials), random modifications (perturbations), and so on by equalizing the number of runs of heuristic algorithms.

4) In the Graph Edit Distance (GED) problem, MDP is designed to modify \mathcal{G}_{1} to be more aligned with \mathcal{G}_{2}. This direction seems different from the other cases where the constraints become worse. How the GED problem fulfills Eq.(2) is not straightforward because constraints in matching two graphs are different from running algorithms on one graph.


Clarity
The manuscript is clearly written and easy to follow.

Significance
If all the technical issues described above are resolved, the proposed approach is worth considering, because the proposed framework can envision the direction that RL can boost the performance of many heuristic or state-of-art algorithms by exploiting such algorithms.

Other weakness
When the three problems are implemented, authors use all different neural network structures. While it is non-trivial to design neural network structure for MDP on graphs, authors do not suggest any guidance or ablation study when using different kinds of network structure.

**Time Spent Reviewing:**

5

---

> ### Author Response · Authors · 2021-08-09
> **Response to Reviewer wSnT**
>
> Thank you for your recognition to our work's novelty, clarity, potential influence and strong empirical performance. We really appreciate your summarization of our key contribution: “apply RL to modify a given (constraint) graph so that a heuristic algorithm over the modified graph can result in the (near-)optimal solution of the original problem”. Your valuable comments are thought-provoking. We elaborate the Proposition and add a random search baseline named Random-BiHyb. We set out below our responses to each of the questions.
>
>
> **Q1: Proposition is unclear**
>
> * RE: In our Proposition, $\mathcal{G}^\*$ is constructed such that **the only feasible solution of $\mathcal{G}^\*$ is $\mathbf{x}^\*$**. Thus, no matter what heuristic algorithm is applied, it will output $\mathbf{x}^*$. Besides, to better clarify our point, we slightly reword the Proposition into:
>
>   **Proposition.** _Suppose the optimal solution $\mathbf{x}^*$ to $\mathcal{G}$ is given, at least one instance of $\mathcal{G}^*$ can be constructed by modifying $\mathcal{G}$, such that $\mathcal{G}^*$ has the only feasible solution of $\mathbf{x}^*$._
>
>   _Proof._ The optimal solution $\mathbf{x}^*$ to $\mathcal{G}$ must satisfy all the constraints in the original graph, $\mathcal{G}$. If $\mathbf{x}^*$ is not the only feasible solution to $\mathcal{G}$, we can add constraints to force that $\mathbf{x}^*$ is the only solution, which is equal to modify $\mathcal{G}$ to construct $\mathcal{G}^*$, such that $\mathbf{x}^*$ is the only feasible solution to $\mathcal{G}^*$.
>
>   For more details, please refer to the section **"We elaborate the Proposition"** in https://openreview.net/forum?id=nwWLJsTJfv&noteId=fWczDIedyzE
>
>
>
> **Q2:** **Compare with other graph modification methods**
>
> * RE: We add a new entry named **Random-BiHyb** which is relatively straightforward and efficient. Random-BiHyb performs random search *instead of PPO* when modifying the graph, and its search steps are equal to PPO-BiHyb.
>
>   **Result analysis.** In summary, Random-BiHyb can improve the lower-level heuristic algorithm, suggesting the effectiveness of the bi-level optimization pipeline. Besides, Random-BiHyb is inferior to PPO-BiHyb, suggesting that PPO is more effective than naive random search.
>
>   For more details, please refer to the section **"We add a new entry Random-BiHyb"** in https://openreview.net/forum?id=nwWLJsTJfv&noteId=fWczDIedyzE
>
>   **The identity baseline** suggested by the reviewer (just more trials) may not work for DAG scheduling and GED, because those heuristic algorithms are deterministic, and repeating more trials cannot provide different and improved results. The LKH in the HCP problem can be initialized with different random seeds, and the LKH3-accu serves as the identity baseline by adding more random trials to LKH3-fast (which is used by PPO-BiHyb and Random-BiHyb). If there are any misunderstandings, feel free to let us know so that we can update the result.
>
>
>
> **Q3:** **Graph modification is likely to make the problem harder**
>
> * RE: **Maybe not. See the new Random-BiHyb results.** The random search method Random-BiHyb suggested by the reviewer outperforms most heuristic baselines, showing that getting a better graph structure is feasible by even random search. Of course, there are chances that the heuristic algorithm produces worse results on modified graphs, but such circumstances are assigned with negative rewards during training, and we prevent the agent from generating worse graphs during inference by local beam search. As we discussed in the Proposition, there must exist an optimal direction that can be achieved by modifying the graph, and this paper aims to develop a reinforcement learning method for that.
>
>   **Hints from traditional optimization.** As discussed in L115-118, it is also a classic idea to modify constraints to facilitate problem-solving. For example, the cutting plane method in integer linear programming [1, 2] proposes to narrow down the search space towards the convex hull of the constraint integer variables, by which our paper is inspired.
>
>   **References:**
>
>   [1] Gomory. *Outline of an algorithm for integer solutions to linear programs and an algorithm for the mixed integer problem*. 50 Years of Integer Programming 1958-2008.
>
>   [2] Tang et al. *Reinforcement learning for integer programming: Learning to cut*. ICML 2020
>
>
>
> **Q4: The constraints of DAG scheduling and HCP become worse, and the constraints of GED become better.**
>
> * RE: **All constraints become better.** For DAG scheduling, the agent learns some important execution orders of jobs to help the heuristic to jump out of local optima. For HCP, the agent also learns to block some critical paths to help the heuristic jump out of local optima. As discussed above in Q3, this paper aims to develop graph modification techniques to make the problem to be better solved by a heuristic algorithm.
>
>
>
> **Q5:** **How to fit GED into Eq (2)**
>
> * RE: We regret that we cannot explain the connection of GED and Eq (2) in detail due to limited pages. Here is our explanation of how to fit GED into Eq (2), and we will update it in the final version.
>
>   $$
>   \\min_{\\mathbf{x}^\\prime, \\mathcal{G}^\\prime_1} f(\\mathbf{x}^\\prime| \\mathcal{G}_1, \\mathcal{G}_2) \qquad s.t. \quad H_j(\\mathcal{G}^\\prime_1, \\mathcal{G}_1) \\leq 0, \text{for}\ j = 1 ... J
>   $$
>   $$
>    \qquad \qquad \qquad \qquad \qquad \qquad  \\mathbf{x}^\\prime \\in \\arg \\min\_{\mathbf{x}^\prime} \\left\\{f(\\mathbf{x}^\\prime| \\mathcal{G}^\\prime_1, \\mathcal{G}_2) :
>   h_i(\\mathbf{x}^\\prime, \\mathcal{G}^\\prime_1, \\mathcal{G}_2) \\leq 0, \\text{for}\\ i = 1 ... I\\right\\}
>   $$
>
>   where $f(\mathbf{x}^\prime| \mathcal{G}_1, \mathcal{G}_2)$ is the upper-level objective: the graph edit cost given $\mathcal{G}_1, \mathcal{G}_2$ and $\mathbf{x}^\prime$. The decision variable $\mathbf{x}^\prime$ encodes how to edit the nodes in $\mathcal{G}_1$ to $\mathcal{G}_2$, and the edge editions can be induced based on node editions. The upper-level constraints $H_j(\mathcal{G}^\prime_1, \mathcal{G}_1) \leq 0$ ensure that $\mathcal{G}^\prime_1$ has at most $K$ modification steps from $\mathcal{G}_1$. The lower-level optimization problem is a GED between $\mathcal{G}_2$ and the modified $\mathcal{G}_1^\prime$, where $f(\mathbf{x}^\prime| \mathcal{G}^\prime_1, \mathcal{G}_2)$ is the lower-level objective, and $h_i(\mathbf{x}^\prime, \mathcal{G}^\prime_1, \mathcal{G}_2) \leq 0$ are the constraints of the GED.
>
>
>
> We hope the above elaborations could help you get through the confusion. We remain available for any further questions you may have and we look forward to your feedback at your earliest convenience.

---

### Official Review · Reviewer_aSHH · 2021-07-19

**Rating:** 3
**Confidence:** 4

**Summary:**

This paper presents an interesting idea to solve Combinatorial Optimization on Graphs by using Reinforcement Learning in order to modify the underlying graph in order to make it easier for a heuristic algorithm. The reward for the RL algorithm is simply the increase found by the heuristic algorithm between the original graph and the modified one. The authors test their algorithm on several real-world dataset related to DAG scheduling, graph edit distance and hamiltonian cycle problem.

**Limitations And Societal Impact:**

OK

**Main Review:**

I like the idea to mix RL with a heuristic algorithm but there are some points which need to be clarified:
1- in each case, a GCN is used in Algorithm 1 to predict P(a_1) and P(a_2|a_1) for adding/removing//modifying edge (a_1,a_2) but I do not understand how the weights of this GCN can be learned? The reward is not a differentiable function?
2- I do not understand how a ResNet block can be used on graphs? On page 5, the authors refer to [20] the Mask R-CNN paper. I do not see the connection, CNN are used on images not graphs?
3- I had a look at the TPC-H dataset but it is not clear at all from the website how this is related to the task. Since the authors are using datasets which are not standard in the ML community, they should explain them and give at least basics statistics.
4- on page 5, the authors say that they use GCN on the graph and on its complement. This is rather surprising as GCN are typically used on sparse graphs. Can they comment on this?
It would have been much better to provide the code associated with the experiments.

**Time Spent Reviewing:**

3

---

> ### Author Response · Authors · 2021-08-06
> **Response to Reviewer aSHH**
>
> We appreciate the reviewer for identifying the novelty of this paper. However, there might be certain misunderstandings towards our paper. We set out below our responses to each of the questions and we remain available for any of your further queries.
>
> **Q1: I do not understand how the weights of this GCN can be learned? The reward is not a differentiable function?**
>
> * RE: The reward is not a differentiable function, and researchers have developed various deep reinforcement learning methods to work with non-differentiable reward functions, including PPO adopted in this paper. We elaborate the PPO objective function described in the paper:
>
>   $$J(\theta) = \min(r_\theta\cdot A, \ clip(r_\theta, 1-\epsilon, 1+\epsilon)\cdot A)$$
>
>   In the following, we omit the "state" in conditional probability for compact illustration. Here $r_\theta=\frac{\pi_\theta(a_1, a_2)}{\pi_{old}(a_1, a_2)}$ is known as the importance sampling ratio, $\pi_\theta(a_1, a_2)$ is the policy neural network parameterized by $\theta$, and $\pi_\theta(a_1, a_2)= P(a_1)P(a_2|a_1)$ where $P(a_1), P(a_2|a_1)$ are action probabilities predicted by GCN and other networks. $\pi_{old}(a_1, a_2)$ is the policy network of the old model that is used to explore the environment. PPO follows the popular actor-critic pipeline, and $A$ is called "advantage", which is the reward subtracted by the prediction of the critic network. As described above, only $A$ is relevant to the reward and it is not required to be differentiable, and gradients are propagated through the policy network $\pi_\theta(a_1, a_2) $. For more details about PPO, please refer to the original paper [5].
>
>   We would like to point out that there exists a family of reinforcement learning combinatorial optimization methods with non-differentiable reward [1, 2, 3, 4].
>   Although limited pages do not allow us to list full details of the reinforcement learning algorithm, we believe our readers are equipped with adequate background knowledge in this regard.
>
> **Q2: I do not understand how a ResNet block can be used on graphs?**
>
> * RE: Here the ResNet block should be read together with L47, where reference is made to the paper [6].
>
>   The ResNet module in this paper is composed of 2 fully connected layers with a residual link, followed by another fully connected layer for output. The input is the embedding vector of each node, and the output is the predicted probability of selecting this node.
>
>   Although ResNet is originally designed for image classification with CNNs, the idea of adding residual links to facilitate the learning process is universal. Also as illustrated in Figure 2 of the ResNet paper [6], the "weight layers" may not necessarily be CNN layers. Besides, the idea of residual links has also been adopted to graph neural networks. Please refer to [7, 8].
>
> **Q3: About the TPC-H dataset**
>
> * RE: The TPC-H dataset contains real-world jobs that are represented by directed acyclic graphs (DAGs). We build a job scheduling task by jointly scheduling 50, 100, 150 jobs sampled from the TPC-H dataset to reflect real-world challenges. TPC-H is also used by previous reinforcement learning papers [1, 4] on computer job scheduling.
>
>   Here are some statistics on TPC-H as per the reviewer’s requirement and we will update the final version accordingly:
>
>   * The DAG in TPC-H dataset has 9.18 nodes in average, and the smallest job has 2 nodes and the largest one has 18 nodes. The average resource requirement is 125.8, and the minimum is 1 and the maximum is 593. The average task duration is 1127.2 sec, and the minimum is 16.3 sec and the maximum is 4964.5 sec.
>
> **Q4: About the complementary graph in DAG scheduling**
>
> * RE: This seems to be a misunderstanding. The graphs in the DAG scheduling problem are directed. In the paper, we mention that "the first GCN processes the original DAG, and the second GCN processes the DAG with all edges reversed". The **"reversed edges"** here means changing the direction of the edges, and the sparsity does not change after reversing the edges.
>
> **Q5: About the code**
>
> * RE: We are truly regretful to say that the code is currently under internal reviews before being made public. Once this paper is accepted, we will publish the code, datasets, and pretrained models.
>
> We hope the above elaborations could help you get through the confusion. We remain available for any further questions you may have and we look forward to your feedback at your earliest convenience.
>
>
>    **References**
>
>    [1] Chen and Tian. *Learning to perform local rewriting for combinatorial optimization.* NeurIPS 2019.
>
>    [2] Dai et al. *Learning Combinatorial Optimization Algorithms over Graphs.* NeurIPS 2017.
>
>    [3] Lu et al. *A Learning-based Iterative Method for Solving Vehicle Routing Problems.* ICLR 2020.
>
>    [4] Mao et al. *Learning scheduling algorithms for data processing clusters.* SIGCOMM 2019.
>
>    [5] Schulman et al. *Proximal Policy Optimization Algorithms*. Axiv 2017.
>
>    [6] He et al. *Deep residual learning for image recognition.* CVPR 2016.
>
>    [7] Li et al. *DeepGCNs: Can GCNs Go As Deep As CNNs?* ICCV 2019.
>
>    [8] Dehmamy et al. *Understanding the Representation Power of Graph Neural Networks in Learning Graph Topology*. NeurIPS 2019.

---

> > ### Comment · Reviewer_aSHH · 2021-08-19
> > **after rebuttal**
> >
> > Thank you for your answer, it cleared up some of my concerns, and I advise you to include discussions about these things in the paper. I still maintain that the presentation itself is not strong enough for acceptance, and not something that is possible to fix before camera-ready.

---

### Author Response · Authors · 2021-08-09
**Thank all reviewers and our response to shared issues**

We would like to express our sincere gratitude to all reviewers for their valuable comments. Here we summarize the shared issues raised by multiple reviewers and provide our responses to them.



## **We elaborate the Proposition**

We understand that the Proposition might be a bit confusing due to limited pages. We slightly reword the proposition to clarify our points and we will better position the Proposition in the final version.



**Proposition.** _Suppose the optimal solution $\mathbf{x}^*$ to $\mathcal{G}$ is given, at least one instance of $\mathcal{G}^*$ can be constructed by modifying $\mathcal{G}$, such that $\mathcal{G}^*$ has the only feasible solution of $\mathbf{x}^*$._


_Proof._ The optimal solution $\mathbf{x}^*$ to $\mathcal{G}$ must satisfy all the constraints in the original graph, $\mathcal{G}$. If $\mathbf{x}^*$ is not the only feasible solution to $\mathcal{G}$, we can add constraints to force that $\mathbf{x}^*$ is the only solution, which is equal to modify $\mathcal{G}$ to construct $\mathcal{G}^*$, such that $\mathbf{x}^*$ is the only feasible solution to $\mathcal{G}^*$.


**Example.** We take _DAG scheduling_ as an example to clarify this Proposition. Without loss of generality, suppose $\mathbf{x}^*$ be the sequence of nodes $1$ to $n$ in order. Correspondingly, we can modify the original graph $\mathcal{G}$ to acquire $\mathcal{G}^*$ to which $\mathbf{x}^*$ is the only feasible solution: if any edge connecting two adjacent nodes in the sequence, e.g. $(i, i+1)$ ($0 < i < n$), does not exist in $\mathcal{G}$, it is added to $\mathcal{G}$. As a result, this exact node sequence from $1$ to $n$, $\mathbf{x}^*$, is the only feasible solution, which will be also the output of any heuristic algorithm. Take another example, if two nodes $i$ and $i+1$ need to be executed in parallel, we can add two edges: $(i-1, i)$ and $(i-1, i+1)$.


**Some Remarks.** This Proposition provides a theoretical grounding of developing graph-modification methods to tackle combinatorial problems. It is worth noting that $\mathcal{G}^*$ only suggests that graph modification is a promising direction and finding $\mathcal{G}^*$ given $\mathcal{G}$ is nearly infeasible in practice. In this paper, we propose to improve the solving quality for heuristic algorithms by finding optimized (not necessarily optimal) graphs by learning based on the bi-level reformulation of the original single-level problem.



## **We add a new entry Random-BiHyb**

Many thanks for the valuable suggestions from the reviewer RkUB and wSnT. We add a new entry named **Random-BiHyb** for all experiments. Random-BiHyb performs random search *instead of PPO* when modifying the graph, and its search steps are equal to PPO-BiHyb.



**Result analysis.** In summary, Random-BiHyb can improve the lower-level heuristic algorithm (in bold), suggesting the effectiveness of the bi-level optimization pipeline. Besides, Random-BiHyb is inferior to PPO-BiHyb, suggesting that PPO is more effective than naive random search.



**DAG-scheduling**

|                   | TPC-H-50 (#nodes=467.2) |                       | TPC-H-100 (#nodes=929.8) |                       | TPC-H-150 (#nodes=1384.5) |                       |
| ----------------- | ----------------------- | --------------------- | ------------------------ | --------------------- | ------------------------- | --------------------- |
|                   | objective $\downarrow$  | relative $\downarrow$ | objective $\downarrow$   | relative $\downarrow$ | objective $\downarrow$    | relative $\downarrow$ |
| **critical path** | 9821                    | 0.0%                  | 16914                    | 0.0%                  | 24429                     | 0.0%                  |
| PPO-Single        | 10578                   | 7.7%                  | 17282                    | 2.2%                  | 24822                     | 1.6%                  |
| **Random-BiHyb**      | 9270                    | -5.6%                 | 15580                    | -7.9%                 | 22930                     | -6.1%                 |
| PPO-BiHyb (ours)  | **8906**                | **-9.3%**             | **15193**                | **-10.2%**            | **22371**                 | **-8.4%**             |



**Graph edit distance**

|                  | AIDS-20/30 (#nodes=22.6) |  | AIDS-30/50 (#nodes=37.9) |  | AIDS-50+ (#nodes=59.6) |                              |
| ---------------- | ------------------------ | ------------------------ | ---------------------- | --------------------- | ---------------------- | --------------------- |
|                  | objective $\downarrow$   | relative $\downarrow$    | objective $\downarrow$ | relative $\downarrow$ | objective $\downarrow$ | relative $\downarrow$ |
| **IPFP**         | 37.4                     | 0.0%                     | 70.4                   | 0.0%                  | 101.9                  | 0.0%                  |
| PPO-Single       | 56.5                     | 51.1%                    | 110.0                  | 56.3%                 | 183.9                  | 80.5%                 |
| **Random-BiHyb**     | 33.1                     | -11.5%                   | 66.0                   | -6.3%                 | 82.4                   | -19.1%                |
| PPO-BiHyb (ours) | **29.1**                 | **-22.2%**               | **61.1**               | **-13.2%**            | **77.0**               | **-24.4%**            |



**Hamiltonian cycle problem**

|                  | FHCP-500/600 (#nodes=535.1) |                         |
| ---------------- | --------------------------- | ----------------------- |
|                  | TSP objective $\downarrow$  | found cycles $\uparrow$ |
| **LKH3-fast**    | 13.8                        | 0%                      |
| LKH3-accu        | **6.3**                         | 20%                     |
| PPO-Single       | 465.7                       | 0%                      |
| **Random-BiHyb**     | 10.0                        | 0%                      |
| PPO-BiHyb (ours) | 8.5                         | **25%**                     |



## **About the code**

We are truly regretful to say that the code is currently under internal reviews before being made public. Once this paper is accepted, we will make every effort to publish the code, datasets, and pretrained models.

---

### Author Response · Authors · 2021-08-31
**Inquiry for post-rebuttal comments**

Dear reviewers,

I would like to express our sincere gratitude for your valuable comments on this paper. I would also like to thank Reviewer aSHH for responding to our rebuttal, although I feel sorry that our response to your questions does not change your mind.

Since the discussion period is approaching its ending, for other reviewers, we would be glad to hear from you about whether our rebuttal has addressed your concerns? If you have any further questions and concerns, feel free to post comments so that we can respond to your questions and concerns.

Especially, I notice that Reviewer RkUB downgrades the score from 5 to 4, may I know if there are any points that still seem unclear? We will truly appreciate it if you could post some comments so that we can improve this paper accordingly.

---

### Decision · Program_Chairs · 2021-09-27

**Decision:**

Accept (Poster)

**Comment:**

This interesting paper explores a bilevel approach to solving combinatorial optimization
problems by using a learning algorithm to edit the graph until a
(non-learning-based) heuristic can return the solution.
The exposition should be improved, experiments should show ablation studies to understand
the effect of network structure, and of course, it would be much better for code to be made available;
but the effort taken by the authors in the review period increases my confidence that
the camera-ready version will address these issues.